# MoGIC: Boosting Motion Generation via Future-aware Behavior Understanding and Visual Context

## Abstract

Existing text-driven motion generation methods primarily focus on bidirectional mapping between language and motion, yet they often struggle to capture the high-level semantic structures and future behavior patterns that govern how actions unfold. Moreover, the absence of visual conditioning limits synthesis accuracy, as language alone cannot specify fine-grained spatiotemporal trajectories or environmental context. We present MoGIC, a unified multimodal framework that jointly models future-aware behavior understanding and multimodal-conditioned motion generation. MoGIC formulates future-behavior predicting as inferring high-level future semantic patterns from partial observations, while leveraging visual priors to resolve ambiguities inherent in text-only conditioning. We further introduce a mixture-of-attention mechanism with adaptive scope to facilitate effective interactions between multimodal tokens and temporal motion segments, thereby mitigating the impact of non-strict timing alignment. To support this paradigm, we curate Mo440H, a 440-hour tri-modal benchmark aggregated from 21 high-quality motion datasets. Extensive experiments demonstrate substantial improvements in generation fidelity and multimodal versatility of MoGIC: (1) a 36% reduction in FID on HumanML3D and Mo440H; (2) superior captioning performance compared to LLM-based methods using only a lightweight text head; (3) capabilities in future-aware behavior prediction and vision-conditioned motion synthesis. Together, these results advance the state of the art in motion understanding and multi-conditioned generation.

## 1 Introduction

Human motion generation has emerged as a central research direction in artificial intelligence, with wide applications in animation, virtual reality, and embodied agents. Recent advances in generative modeling have enabled text-conditioned methods to synthesize increasingly realistic motions from natural language descriptions. Despite this progress, current systems remain far from capturing the full complexity, and semantic structure of human motion.

Most existing methods Guo et al. (2024); Meng et al. (2025) focus on learning direct correspondences between language and motion without explicitly modeling motion understanding, whereas recent GPT-based approaches Jiang et al. (2023a); Wu et al. (2025a); Luo et al. (2024); Jiang et al. (2024) extend this paradigm by treating discretized motion as a second language, thereby enabling bidirectional generation between motion and text. However, such bidirectional modeling reduces the motion-language association to a cross-modal mapping and remains limited in capturing the semantic structures and behavior patterns that govern how actions will unfold over time. Consequently, despite leveraging large-scale linguistic priors, these models remain less effective than specialized motion generation frameworks. Intuitively, a model should not only understand how actions are performed. At a higher semantic level, a model should also learn the structural dependency between early motion and logically consistent future behaviors.

Another critical limitation lies in the absence of visual modality, which leads to motions that lack precision and personalization. Language alone is inherently ambiguous and cannot fully specify fine-grained spatiotemporal trajectories or environmental context; even when such content are ex-

plicitly described, their symbolic form does not readily translate into precise generation signals. For instance, the instruction "a person walks forward, turns left, then picks up an apple" leaves unspecified the walking distance, turning angle, and spatial location of the apple, forcing the model to sample from a highly uncertain distribution and often producing unsatisfactory results. By contrast, visual inputs naturally provide joint trajectories and environmental context, such as human-scene or human-object interactions. Without such visual grounding, models fail to exploit these priors, resulting in less accurate motions and restricting their applicability to text conditioned generation.

We introduce MoGIC, a unified framework that integrates multi-conditioned generation with future-aware behavior understanding. The key insight is that early motion segments contain latent cues about how actions are likely to unfold, and that such cues must be captured through structured semantic modeling rather than direct language-motion mapping. Accordingly, MoGIC employs a disentangled design in which high-level behavior semantics and low-level motion trajectories are handled by separate generation heads. The understanding head predicts the discrete behavior description of a complete sequence given partial observations, while the motion head synthesizes continuous motion sequences under multimodal conditions. This division of roles enables the model to jointly optimize future-aware behavior understanding and motion synthesis without conflating symbolic semantics with continuous kinematics. This design yields substantial improvements in motion quality, achieving a 35.2% reduction in FID compared to training without predicting behavior semantics in MoGIC.

To further address the ambiguity of purely textual conditioning, we incorporate visual modality as an additional condition. Unlike pose estimation methods that reconstruct dense trajectories, we use low-frame-rate image sequences as weak but informative priors. These visual conditions introduce auxiliary perceptual priors into the training process, thereby enhancing the quality of representation learning and yielding consistent improvements across downstream tasks. Furthermore, vision provides conditions that alleviate the inherent ambiguity of language, enabling the generation of motions that are more precise and controllable. Moreover, the integration of vision unlocks the model's capacity to handle a wider range of tasks beyond text-only conditioning, such as vision-conditioned motion completion or generating diverse motions from sparse video frame.

Our video inputs (sampled at 1 fps) are not temporally aligned with motion sequences (30 fps). Instead of enforcing rigid frame-to-frame correspondence, we leverage them as priors for trajectory and scene context. Similarly, text only partially aligns with motion, as some phrases correspond to specific fragments. To handle such partial correspondences, we introduce a mixture-of-attention mechanism with adaptive scope, enabling motion tokens to interact with the most relevant conditional tokens across granularities. This strengthens local-global alignment and mitigates confusion from temporal mismatches or ambiguous conditions.

We adopt a two-stage training strategy. In the first stage, MoGIC learns cross-modal consistency by jointly optimizing multimodal-conditioned motion generation loss and future-aware behavior understanding loss, enabling robust generalization across different input types. In the second stage, we finetune MoGIC on task-specific objectives such as language-to-motion generation, effectively transferring visual priors and behavior understanding capabilities to downstream tasks.

To enable tri-modal learning, we curate and automatically annotate 21 high-quality motion datasets into a large-scale benchmark, Mo440H, which spans 440 hours of single-person motions, human-human interactions, and human-object interactions. Extensive experiments on HumanML3D and Mo440H demonstrate the versatility of MoGIC as a unified motion generation framework capable of handling multiple conditioning modalities and tasks. On the language-to-motion task, MoGIC achieves significant improvements, reducing FID by 38.6% and 34.6% respectively. On motion captioning (with fine-tuning) and future-aware behavior understanding (without fine-tuning), its lightweight text-generation heads outperform LLM-based baselines despite using far fewer parameters. Furthermore, incorporating the visual modality not only enhances controllability and generation fidelity, but also enables new capabilities such as image-to-motion synthesis and vision-conditioned motion completion.

## 2 RELATED WORK

**Motion Generation**  Recent work on text-conditioned motion generation has primarily relied on probabilistic generative models, including GANs Harvey et al. (2020); Ghosh et al. (2021), VAEs Petrovich et al. (2022), and diffusion methods Du et al. (2023); Chen et al. (2023); Shafir et al. (2024); Tevet et al. (2023); Zhang et al. (2023c;b; 2024b); Xie et al. (2024); Zhou et al. (2024); Meng et al. (2025), which generate realistic motion by sampling and refining noise. In parallel, discrete-token approaches Du et al. (2023); Zhong et al. (2023); Zhang et al. (2023a); Pinyoanun-tapong et al. (2024a); Shi et al. (2025); Wang et al. (2025); Jeong et al. (2025) use vector-quantized autoencoders to construct a motion vocabulary, with transformers modeling token sequences either autoregressively or through masked denoising.

More recent studies have begun to explore the use of large language models to capture the rich temporal and semantic structure of human motion Jiang et al. (2023a); Luo et al. (2024); Liang et al. (2024a); Wang et al. (2024); Wu et al. (2025b). By treating motion as a discretized foreign language, these methods learn bidirectional mappings between text-to-motion (T2M) and motion-to-text (M2T). However, such mappings remain at the level of superficial pattern matching, and the potential of bidirectional alignment to improve motion generation has not been fully explored. In this paper, we advance motion generation by enabling the model to infer the underlying causes of motion.

More modalities have also been explored as conditions. For instance, Chen et al. (2025) unify audio and text representations of 3D human motions using LLMs, while MotionAnything Zhang et al. (2025) combines text and music to generate more controllable dance motions. Such modalities provide complementary information, yet the crucial role of visual inputs in motion generation remains underexplored. In this paper, we incorporate visual modality to reduce the ambiguity of text-only descriptions and extend the model's applicability to a wider range of downstream tasks.

**Motion Dataset**  In recent years, numerous motion datasets have emerged to capture diverse human activities. AMASS Mahmood et al. (2019) provides high-quality 3D mocap data, further extended by BABEL Punnakkal et al. (2021) with segment-level categorical labels and HumanML3D Guo et al. (2022) with sequence-level textual descriptions. Beyond these, many datasets Mehta et al. (2017); Fieraru et al. (2021); Cai et al. (2022); Xiong et al. (2024); Liu et al. (2022); Tripathi et al. (2023) capture a broad spectrum of movements ranging from daily actions to gestures and yoga. EgoBody Zhang et al. (2022) and InterGen Liang et al. (2024b) focus on two-person interactions, while others Lv et al. (2025); Zhao et al. (2024); Li et al. (2023); Jiang et al. (2023b); Taheri et al. (2020) emphasize human–object interactions. Collaborative and scene-aware scenarios are also addressed by datasets such as CORE4D Liu et al. (2024b), Humanise Wang et al. (2022), PROX Hassan et al. (2019), and HOI-M3 Zhang et al. (2024a).

Recent works Lin et al. (2023); Lu et al. (2025); Fan et al. (2025) extract motions from online videos via pose estimation. However, this approach results in lower fidelity and exhibits content bias. For instance, MotionMillion Fan et al. (2025) contains more than 70% sports-related activities such as martial arts, fitness, and dance. In this paper, we integrate and re-annotate over twenty high-quality motion datasets, achieving large-scale coverage while avoiding the quality issues and content bias of video-derived data.

## 3 MODEL ARCHITECTURE

We propose MoGIC, a unified framework for future-aware behavior understanding and human motion generation conditioned on multimodal inputs, including language, vision, and partially visible motion sequences. As shown in Figure 1, each modality is first projected into the latent space via modality-specific encoders, with random masking applied to motion tokens for generative masked modeling. A Conditional Masked Transformer (CMT) then integrates the projected conditioning signals at both global-level and fine-grained conditions to modulate the masked motion tokens. The resulting motion tokens serve as a unified representation that generates both high-level behavior descriptions and complete motion latent sequences, which are subsequently reconstructed into the original motion domain through a motion decoder.

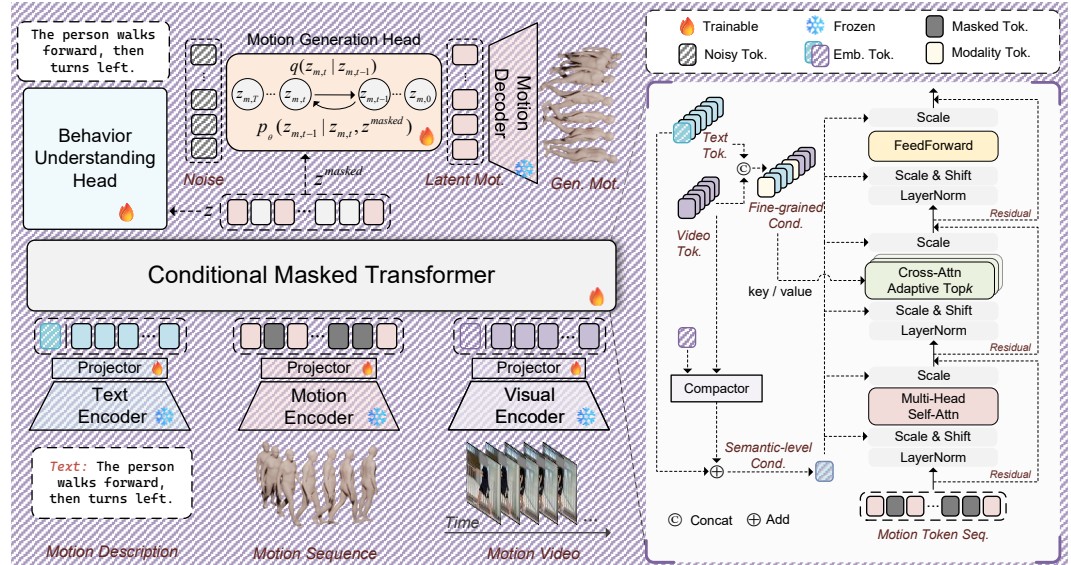

Figure 1: Overview of MoGIC. The framework consists of modality-specific encoders, a Conditional Masked Transformer (CMT), a Motion Generation Head (MGH), and a Behavior Understanding Head (BUH). Language, vision, and motion inputs are first processed by their respective encoders to produce latent tokens. Motion tokens are randomly masked and passed through the CMT, where global-level and fine-grained conditions modulate the motion token in series. The resulting conditional tokens $z$ are used in two branches: (i) the masked motion tokens are reconstructed via the MGH, which denoises them into clean motion latent tokens and decodes them into motion sequences; (ii) $z$ serves as key and query signals for the SPH to predict the behavior description.

## 3.1 MODALITY-SPECIFIC ENCODER

**Motion/Language/Vision Encoder**    Given a motion sequence $\boldsymbol{x}_m \in \mathbb{R}^{l_m \times c_m}$ with $l_m$ frames and $c_m$ feature dimensions, we adopt a temporal convolutional auto-encoder to project motion into a compact latent space $\boldsymbol{z}_m = f_m(\boldsymbol{x}_m) \in \mathbb{R}^{l'_m \times d_m}$, where $l'_m \leq l_m$ and $d_m$ is the motion embedding dimension. A symmetric decoder reconstructs the motion as $\hat{\boldsymbol{x}}_m = g_m(\boldsymbol{z}_m)$, and the auto-encoder is trained with a smooth L1 reconstruction loss $\mathcal{L}_{\text{rec}} = \text{SmoothL1}(\hat{\boldsymbol{x}}_m, \boldsymbol{x}_m)$. For text, a frozen CLIP encoder outputs token-level embeddings $\boldsymbol{z}_t = f_t(\boldsymbol{x}_t) \in \mathbb{R}^{l_t \times d_t}$, where the [CLS] token $\boldsymbol{z}_t^g$ captures global semantics. For vision, video frames sampled at 1 fps are encoded as $\boldsymbol{z}_v^p = f_v(\boldsymbol{x}_v^p) \in \mathbb{R}^{d_v}$, and aggregated into a global representation by attention layer $\boldsymbol{z}_v^g = \text{attn}(\boldsymbol{q}_v, \boldsymbol{z}_v, \boldsymbol{z}_v)$ with a learnable query vector $\boldsymbol{q}_v$.

## 3.2 CONDITIONAL MASKED TRANSFORMER

The Conditional Masked Transformer integrates multimodal conditioning signals into motion tokens through two operations: (i) global-level modulation, which injects fused text–vision context into the motion representation via adaptive normalization to ensure pattern consistency, and (ii) mixture-of-attention with adaptive scope, realized through the adaptive Top-$k$ cross-attention mechanism that dynamically aligns motion tokens with the most relevant text-vision snippets while adaptively determining the scope of attended context. At each layer of the CMT, motion tokens first pass through a self-attention module, then receive fine-grained conditions via the adaptive Top-$k$ cross-attention, and finally go through a feed-forward network to produce the output representation.

**Global-level Modulation**    We adopt adaptive LayerNorm modulation. The global multimodal context vector $\boldsymbol{c}^g = \boldsymbol{z}_t^g + \boldsymbol{z}_v^g$ is mapped to modulation coefficients $(\alpha_c, \beta_c, \gamma_c) = W_{\text{ada}}(\boldsymbol{c}^g) \in \mathbb{R}^{d_m}$ via a lightweight MLP. Given a normalized motion token $\bar{\boldsymbol{z}}_m = \text{LN}(\boldsymbol{z}_m)$, modulation and gated residual connection are applied as

$$\boldsymbol{z}_m \leftarrow \boldsymbol{z}_m + \gamma_c \odot h(\alpha_c \odot \bar{\boldsymbol{z}}_m + \beta_c) \tag{1}$$

where $h(\cdot)$ denotes the corresponding sub-layer transformation (self-attention, cross-attention, and feed-forward layer in CMT). This formulation ensures that global multimodal context consistently modulates motion representation while flexibly controlling residual pathways.

**Mixture-of-Attention with Adaptive Scope** To enable fine-grained dynamic alignment between motion tokens and multimodal conditioning signals, we employ the mixture-of-attention mechanism that operates on concatenated token-level condition embeddings $\boldsymbol{c}^{tok} = [\boldsymbol{z}_v; \boldsymbol{z}_t] \in \mathbb{R}^{(p+l) \times d}$. When a modality is missing, its slot in $\boldsymbol{c}^{tok}$ is replaced by a learnable embedding. Given query motion tokens $\boldsymbol{z}_m \in \mathbb{R}^{l'_m \times d}$, each expert computes queries, keys, and values as $\boldsymbol{q}^e = W_q^e \boldsymbol{z}_m, \boldsymbol{k}^e = W_k^e \boldsymbol{c}^{tok}$, and $\boldsymbol{v}^e = W_v^e \boldsymbol{c}^{tok}$, followed by attention to produce the score matrix $\boldsymbol{A}^e$. To control the effective context scope, we sort $\boldsymbol{A}^e$ in descending order per query and accumulate until a cumulative mass $\tau$ is reached:

$$k_{\text{dyn}}^e = \min \Big( \max \big( \arg \min_k \sum_{j=1}^k \boldsymbol{A}_{(j)}^e \geq \tau, k_{\min}^e \big), k_{\max}^e \Big) \tag{2}$$

where $\boldsymbol{A}_{(j)}^e$ denotes the $j^{th}$ largest weight. The final attention distribution is restricted to the top-$k_{\text{dyn}}^e$ entries. The final output is the sum of all expert contributions

$$\boldsymbol{z} = \sum_{e=1}^E \tilde{\boldsymbol{A}}^e \boldsymbol{v}^e, \quad \tilde{\boldsymbol{A}}_{i,j}^e = \frac{\boldsymbol{A}_{i,j}^e \cdot \mathbf{1} j \in \text{top-}k_{\text{dyn}}^e(i)}{\sum_{j' \in \text{Top-}k_{\text{dyn}}(i)} \boldsymbol{A}_{i,j'}^e} \tag{3}$$

where top-$k_{\text{dyn}}^e(i)$ denotes the set of indices of the attention weights for the $i^{th}$ motion token. This adaptive mixture-of-attention design ensures that motion tokens selectively attend to the most relative condition tokens, while maintaining flexibility to balance pattern consistency and fine-grained alignment across diverse contexts.

## 3.3 DISENTANGLE GENERATION HEAD

Behavior understanding and motion represent two fundamentally different data formats, with the former being linguistically oriented and motion encoding continuous dynamics. To capture this distinction, we adopt disentangled generation heads that separately model the two modalities

**Behavior Understanding Head (BUH)** The Behavior Understanding Head (BUH) aims to capture the complete behavior pattern from the partial observation in textual form. It employs a T5-style Raffel et al. (2020) decoder that, conditioned on the embedding $\boldsymbol{z}$ from the conditional masked transformer, generates the behavior description in an autoregressive manner. Each decoder layer combines self-attention over the partially generated sequence with cross-attention conditioned on $\boldsymbol{z}$

**Motion Generation Head (MGH)** Since motions are continuous, we employ a continuous-time interpolant model in the Motion Generation Head (MGH) following SiT Ma et al. (2024), conditioned on the masked embedding $\boldsymbol{z}^{mask}$. The interpolant at time $t \in [0, 1]$ is defined as:

$$\boldsymbol{z}_{m,t} = \alpha_t \boldsymbol{z}_{m,0} + \sigma_t \boldsymbol{\epsilon}, \quad \boldsymbol{\epsilon} \sim \mathcal{N}(0, I)$$
$$\boldsymbol{v}_\theta(\boldsymbol{z}_{m,t}, t, \boldsymbol{z}^{mask}) \approx \dot{\alpha}_t \boldsymbol{z}_{m,0} + \dot{\sigma}_t \boldsymbol{\epsilon} \tag{4}$$

where $\boldsymbol{z}_{m,0}$ is the latent motion of ground truth and $\boldsymbol{v}_\theta$ predicts the velocity field under conditioning. For sampling, given $\boldsymbol{z}^{mask}$, we integrate the learned velocity field backward in time using either the probability flow ODE or reverse-time SDE:

$$\boldsymbol{z}_{m,0} = \text{Solver}(\boldsymbol{z}_{m,T}, \boldsymbol{v}_\theta, \boldsymbol{z}^{mask}), \quad \boldsymbol{z}_{m,T} \sim \mathcal{N}(0, I). \tag{5}$$

The denoised latent representation $\boldsymbol{z}_{m,0}$ is subsequently provided to the motion decoder to reconstruct the full motion sequence $\hat{\boldsymbol{x}}_m$.

## 4 CROSS-MODAL GENERATIVE TRAINING

We train MoGIC on five tasks: (1) language-to-motion (L2M), which generates motion from textual descriptions; (2) vision-language-to-motion (VL2M), which integrates visual and textual inputs to produce more controllable motion; (3) vision-to-motion (V2M), which synthesizes motion

Table 1: The quantitative results of L2M on HumanML3D. The best results are displayed in bold. Noting that the metric values of some methods are adopted from MARDM (Meng et al., 2025).

| Methods | R Precision↑ | | | FID↓ | Matching↓ | CLIP-score ↑ |
|---|---|---|---|---|---|---|
| | Top 1 | Top 2 | Top 3 | | | |
| T2M-GPT (Zhang et al., 2023a) | $0.470^{\pm.003}$ | $0.659^{\pm.002}$ | $0.758^{\pm.002}$ | $0.335^{\pm.003}$ | $3.505^{\pm.017}$ | $0.607^{\pm.005}$ |
| ReMoDiffuse (Zhang et al., 2023b) | $0.468^{\pm.003}$ | $0.653^{\pm.003}$ | $0.754^{\pm.005}$ | $0.883^{\pm.021}$ | $3.414^{\pm.020}$ | $0.621^{\pm.003}$ |
| MDM-50Step (Tevet et al., 2023) | $0.440^{\pm.007}$ | $0.636^{\pm.006}$ | $0.742^{\pm.004}$ | $0.518^{\pm.032}$ | $3.640^{\pm.028}$ | $0.578^{\pm.003}$ |
| MLD (Chen et al., 2023) | $0.461^{\pm.004}$ | $0.651^{\pm.004}$ | $0.750^{\pm.003}$ | $0.431^{\pm.014}$ | $3.445^{\pm.019}$ | $0.610^{\pm.003}$ |
| MMM (Pinyoanuntapong et al., 2024b) | $0.487^{\pm.003}$ | $0.683^{\pm.002}$ | $0.782^{\pm.001}$ | $0.132^{\pm.004}$ | $3.359^{\pm.009}$ | $0.635^{\pm.003}$ |
| MoMask (Guo et al., 2024) | $0.469^{\pm.004}$ | $0.687^{\pm.003}$ | $0.786^{\pm.003}$ | $0.116^{\pm.006}$ | $3.353^{\pm.010}$ | $0.637^{\pm.003}$ |
| MotionDiffuse (Zhang et al., 2024b) | $0.450^{\pm.006}$ | $0.641^{\pm.005}$ | $0.753^{\pm.005}$ | $0.778^{\pm.005}$ | $3.490^{\pm.023}$ | $0.606^{\pm.004}$ |
| MARDM-DDPM (Meng et al., 2025) | $0.492^{\pm.006}$ | $0.690^{\pm.005}$ | $0.790^{\pm.005}$ | $0.116^{\pm.004}$ | $3.349^{\pm.010}$ | $0.637^{\pm.005}$ |
| MARDM-SiT (Meng et al., 2025) | $0.500^{\pm.004}$ | $0.695^{\pm.003}$ | $0.795^{\pm.003}$ | $0.114^{\pm.007}$ | $3.270^{\pm.009}$ | $0.642^{\pm.002}$ |
| MotionAgent Wu et al. (2025b) | $0.485^{\pm.003}$ | $0.680^{\pm.003}$ | $0.780^{\pm.002}$ | $0.202^{\pm.009}$ | $3.327^{\pm.009}$ | $0.634^{\pm.003}$ |
| MoGIC (ours) *w/o Und. loss* | $0.533^{\pm0.012}$ | $0.731^{\pm0.010}$ | $0.826^{\pm0.010}$ | $0.108^{\pm0.023}$ | $3.078^{\pm0.037}$ | $0.658^{\pm0.001}$ |
| MoGIC (ours) | $\mathbf{0.545^{\pm0.003}}$ | $\mathbf{0.741^{\pm0.003}}$ | $\mathbf{0.835^{\pm0.002}}$ | $\mathbf{0.070^{\pm0.004}}$ | $\mathbf{2.999^{\pm0.011}}$ | $\mathbf{0.669^{\pm0.001}}$ |

Table 2: The quantitative results of L2M on Mo440H-ML. The best results are displayed in bold.

| Methods | R Precision↑ | | | FID↓ | Matching↓ | Diversity↑ |
|---|---|---|---|---|---|---|
| | Top 1 | Top 2 | Top 3 | | | |
| MotionDiffuse (Zhang et al., 2024b) | $0.550^{\pm.001}$ | $0.735^{\pm.001}$ | $0.801^{\pm.002}$ | $0.957^{\pm.010}$ | $2.990^{\pm.007}$ | $12.009^{\pm.104}$ |
| MMM (Pinyoanuntapong et al., 2024b) | $0.601^{\pm.001}$ | $0.798^{\pm.001}$ | $0.887^{\pm.001}$ | $0.237^{\pm.004}$ | $2.420^{\pm.004}$ | $11.883^{\pm.089}$ |
| MoMask (Guo et al., 2024) | $0.610^{\pm.001}$ | $0.801^{\pm.002}$ | $0.886^{\pm.001}$ | $0.205^{\pm.006}$ | $2.353^{\pm.003}$ | $11.963^{\pm.077}$ |
| MARDM-DDPM Meng et al. (2025) | $0.573^{\pm.001}$ | $0.785^{\pm.002}$ | $0.885^{\pm.002}$ | $0.431^{\pm.004}$ | $2.166^{\pm.005}$ | $\mathbf{12.630^{\pm.079}}$ |
| MARDM-SiT Meng et al. (2025) | $0.613^{\pm.001}$ | $0.820^{\pm.002}$ | $0.906^{\pm.001}$ | $0.231^{\pm.003}$ | $2.420^{\pm.005}$ | $12.112^{\pm.079}$ |
| MG-MotionLLM Wu et al. (2025a) | $0.556^{\pm0.002}$ | $0.737^{\pm0.002}$ | $0.834^{\pm0.002}$ | $0.624^{\pm0.008}$ | $2.544^{\pm0.006}$ | $12.252^{\pm0.099}$ |
| MoGIC (ours) *only L2M loss* | $0.637^{\pm0.001}$ | $0.836^{\pm0.001}$ | $0.908^{\pm0.002}$ | $0.201^{\pm0.001}$ | $2.003^{\pm0.007}$ | $12.392^{\pm0.084}$ |
| MoGIC (ours) *L2M + Und. loss* | $\mathbf{0.652^{\pm0.001}}$ | $\mathbf{0.851^{\pm0.001}}$ | $\mathbf{0.926^{\pm0.001}}$ | $0.134^{\pm0.001}$ | $\mathbf{1.889^{\pm0.005}}$ | $12.434^{\pm0.087}$ |
| MoGIC (ours) *L2M + Caption loss* | $0.646^{\pm0.001}$ | $0.845^{\pm0.001}$ | $0.919^{\pm0.001}$ | $0.198^{\pm0.001}$ | $1.910^{\pm0.005}$ | $12.623^{\pm0.090}$ |
| MoGIC (ours) | $0.643^{\pm0.001}$ | $0.844^{\pm0.002}$ | $0.917^{\pm0.002}$ | $0.185^{\pm0.002}$ | $1.915^{\pm0.004}$ | $12.516^{\pm0.077}$ |
| MoGIC (ours) *w/ L2M FT* | $0.651^{\pm0.001}$ | $0.849^{\pm0.001}$ | $0.924^{\pm0.002}$ | $\mathbf{0.123^{\pm0.001}}$ | $1.903^{\pm0.006}$ | $12.511^{\pm0.091}$ |

purely from visual sequences; (4) motion-to-motion (M2M), which reconstructs complete motion from partially observed sequences; and (5) future-aware behavior understanding, which infers high-level motivational factors behind motion. All tasks share a Conditional Masked Transformer with modality-specific conditioning. Motion sequences are encoded into latent tokens $z_m \in \mathbb{R}^{l'_m \times d_m}$, where a subset is randomly masked with learnable tokens for generative reconstruction, and for future-aware behavior understanding, the latter 50% of tokens are additionally truncated. The fused masked sequence and modalities yield a motion embedding $z$, which conditions both the SPH and MGH for behavior understanding and motion generation. Training is driven by a joint loss combining a diffusion-based velocity matching objective for motion and an autoregressive cross-entropy for description:

$$\mathcal{L} = \lambda_{\text{motion}} \, \mathbb{E}_{t,\epsilon} \left[ \left\| v_\theta(z_{m,t}, t, z) - (\dot{\alpha}_t z_{m,0} + \dot{\sigma}_t \epsilon) \right\|_2^2 \right] + \lambda_{\text{und}} \, \mathbb{E}_{(y,z)} \left[ -\sum_{i=1}^{T} \log P(y_i | y_{<i}, z) \right] \quad (6)$$

This unified training framework enables the model to learn a shared latent space where motion generation and behavior understanding are jointly optimized. The decoupled generation paradigm guides the model to capture the underlying motivational factors of motion, while mitigating the semantic entanglement between discrete text and continuous motion representations.

## 5 EXPERIMENTS

### 5.1 INTEGRATED MOTION DATASET

**Motion Dataset** We curated and processed 21 high-quality motion datasets covering diverse scenarios such as single-person activities, human-human interactions, and human-object interactions. All motions were standardized to a 22-joint format, resampled to 30 fps, and capped at 10 seconds. For datasets without textual annotations but with visual modalities, we used Qwen2.5-VL-Max Bai et al. (2025) to generate captions and manually filtered inadequate samples; for those

lacking RGB videos, rendered mesh sequences were adopted instead, with all videos downsampled to 1 fps. The final collection, termed Mo440H, comprises about 440 hours of motion (about 50M frames), 210k textual descriptions, and 140k image sequences. Depending on available modalities, we further organize it into three subsets: Mo440H-All (the whole dataset, for auto-encoder training and cross-modal generative training), Mo440H-ML (motion-language pairs, for language-to-motion and motion-to-language), and Mo440H-MLV (motion-language-vision triplets, enabling visually conditioned tasks).

In addition, we evaluate on the HumanML3D Guo et al. (2022) dataset, a widely used benchmark with about 14k motion sequences and 45k text annotations, following established protocols Meng et al. (2025) for fair comparison with previous work.

**Motion Representation**  We adopt a compact motion representation by removing redundant features (e.g., 6D rotations), following Meng et al. (2025), to mitigate distribution mismatch and generation errors. The motion data is represented as $\boldsymbol{x}_m^i = [\dot{r}^a, \dot{r}^{xz}, \dot{r}^h, j^p]$ at time step $i$, consisting of root angular velocity $\dot{r}^a$, root linear velocities $\dot{r}^{xz}$ in the XZ-plane, root height $\dot{r}^h$, and local joint positions $j^p \in \mathbb{R}^{3(N_j-1)}$, which jointly encode the essential kinematic information for motion.

## 5.2 EXPERIMENT SETTINGS AND EVALUATION METRICS

**Experiment Settings**  All experiments are conducted on RTX4090 GPUs with a batch size of 64 using the Adam optimizer (lr=2e-4, 2000-step warm-up), training for 500 epochs on HumanML3D and 10M iterations on Hu440H ($\approx$40 GB GPU memory). The motion generation loss is optimized every epoch, while the future-aware behavior understanding loss is updated every 4 epochs. The Conditional Masked Transformer (384 channels) uses 1 layer for HumanML3D and 2 layers for Hu440H dataset. Cross-attention employs two parallel modules ($k \in [1,6]$, threshold 0.8; and $k \in [0, \infty]$, threshold 1). The semantic prediction head is a 3-layer T5-style decoder, and the motion head is a diffusion model with a 10-layer MLP (1280 channels).

## 5.3 DOWNSTREAM APPLICATIONS

Following cross-modal generative training, MoGIC supports arbitrary multimodal inputs (language, vision, motion) to produce unified outputs in motion sequences and text descriptions. Further finetuning on specific tasks enhances performance in specialized settings. We evaluate on HumanML3D Guo et al. (2022) and our integrated dataset Mo440H. For HumanML3D, we adopt evaluators from prior work Meng et al. (2025). For the integrated dataset, we train an evaluator on Mo440H following the previous methods Guo et al. (2022).

**Motion Generation and Caption**  We evaluate language-to-motion generation both with and without fine-tuning, as well as motion captioning after finetuning. Experiments are conducted on HumanML3D and Mo440H-ML.

For motion generation on HumanML3D, we adopt a single-stage training strategy, jointly optimizing the motion generation loss and the future-aware behavior understanding loss, achieving substantial improvements over state-of-the-art methods in terms of FID and R-Precision (Tables 1). Results on Mo440H are shown in Tables 2, where MoGIC denotes the model

Table 3: Comparisons of motion in-between tasks on Mo440H-ML. Each setting reports R-precision top 3 (R@3), FID, and Matching score (Match).

| Task | Method | w/o language | | | w/ language | | |
|------|--------|------|------|------|------|------|------|
| | | R@3↑ | FID↓ | Match↓ | R@3↑ | FID↓ | Match↓ |
| pref. | MARDM | 0.874 | 0.286 | 2.808 | 0.912 | 0.194 | 1.972 |
| | MoGIC | 0.892 | 0.173 | 2.172 | 0.943 | 0.128 | 1.644 |
| suff. | MARDM | 0.894 | 0.239 | 2.334 | 0.912 | 0.188 | 1.989 |
| | MoGIC | 0.912 | 0.140 | 1.938 | 0.941 | 0.091 | 1.647 |
| inf. | MARDM | 0.907 | 0.211 | 2.249 | 0.913 | 0.186 | 1.984 |
| | MoGIC | 0.926 | 0.124 | 1.789 | 0.943 | 0.113 | 1.619 |
| circ. | MARDM | 0.896 | 0.249 | 2.358 | 0.913 | 0.175 | 1.980 |
| | MoGIC | 0.912 | 0.147 | 1.979 | 0.943 | 0.109 | 1.639 |

Table 4: Text generation metrics on the test set.

| | | BLEU@1↑ | BLEU@4↑ | ROUGE↑ | BERTScore↑ |
|------|--------|---------|---------|--------|------------|
| H3D | TM2T | 48.90 | 8.27 | 38.1 | 32.2 |
| | MotionGPT | 48.20 | 12.47 | 37.4 | 32.4 |
| | MotionChain | 48.10 | **12.56** | 33.9 | 36.9 |
| | MotionGPT3 | 51.06 | 8.43 | 38.7 | 32.0 |
| | MG-MotionLLM | – | 8.06 | – | 36.7 |
| | MoGIC (ours) | **53.13** | 10.36 | **40.6** | **40.7** |
| Mo440H | T2MT | 28.99 | 15.37 | 36.22 | 29.01 |
| | MG-MotionLLM | 35.47 | 17.97 | 39.07 | 30.95 |
| | MoGIC (ours) | **42.52** | **20.32** | **39.31** | **31.96** |

trained solely through cross-modal generative learning, and MoGIC *w/ FT* represents the variant further fine-tuned on the language-to-motion task. We also present the results without computing the generation loss conditioned on the visual modality (denoted as MoGIC *T2M + Und. loss* in the Table 2). As shown, language-based motion generation achieves better results, but its functionality remains limited. All evaluations are conducted using our retrained evaluator on the Mo440H dataset, following the same protocol as previous work Guo et al. (2022).

In addition, Table 4 reports results for fine-tuning on motion caption task. During fine-tuning, we feed the entire motion sequence as input and generate textual descriptions. Compared with LLM-based methods, the Semantic Prediction Head (SPH) in MoGIC is highly lightweight and does not rely on pre-trained language models, yet it still delivers competitive and effective performance.

**Motion In-Between**  We evaluate our method on the motion in-between task, which generates plausible transitions from partial motion contexts. We consider prefix, suffix, infix, and circumfix completion, predicting missing segments at the beginning, end, middle, or both ends of a motion sequence. Experiments on HumanML3D and Mo440H, compared with MARDM Meng et al. (2025), are reported under two settings: (i) in-between with language, using both textual descriptions and visible motion fragments, and (ii) in-between without language, using only motion fragments. Without task-specific fine-tuning, our method consistently outperforms baselines, as shown in Table 3.

**Future-aware Behavior Understanding**  The future-aware behavior understanding task requires the model to infer the underlying semantic structure and behavioral patterns in text format. Given the first 50% of a motion sequence, the model outputs a complete language description that conveys the underlying behavior patterns. Meanwhile, MoGIC can also generate a future motion sequence aligned with this description. We train two baselines separately for future-aware behavior understanding Wu et al. (2025a) and future motion generation Meng et al. (2025). Without fine-tuning, our model surpasses both, achieving higher quality in understanding and lower FID for the synchronously generated motion continuation, as shown in Figure 2.

**Vision-Augmented Tasks**  We further extend our framework to vision-augmented scenarios, where image sequences serve as additional conditions for motion generation. We focus on two representative tasks: (i) vision-language-to-motion, where textual descriptions and visual frames jointly guide motion synthesis, and (ii) vision-based motion in-between, where visual cues complement partial motion fragments to complete missing segments. These tasks provide a natural and accessible source of conditioning signals that enrich the controllability of generated motions. As shown in Figure 3, when generating a weightlifting motion conditioned only on the text prompt "lift weight by extending legs and back, raising arms", the description neither specifies the exact position of the barbell nor provides the model with a prior about the abstract concept of weight. As a result, the model produces an unrealistic sequence in which the barbell is lifted overhead, which is clearly inconsistent with real-world biomechanics. By incorporating visual modality, however, the model gains explicit information about the barbell's position relative to the body, allowing it to generate natural lifting motions that adhere to realistic constraints.

## 5.4 Ablation Study

**Effectiveness of Future-aware Behavior Understanding** Ablation results on HumanML3D (MoGIC *w/o Und. loss* in Table 1) and Mo440H (MoGIC *only T2M loss* in Table 2) show that removing future-aware behavior understanding task consistently lowers performance, with the largest drops in FID ($-35.2\%$ on HumanML3D, $-33.3\%$ on Mo440H) and retrieval precision ($-0.9\%$ on HumanML3D, $-1.8\%$ on Mo440H). We further replace the understanding loss with a captioning loss, training the model to generate descriptions from complete motion sequences (MoGIC *L2M + Caption loss*). Caption supervision improves motion quality, but the gains are notably smaller than those from future-aware behavior understanding. This underscores that learning the semantic structures and understanding how motion will unfold is crucial for producing high-quality motion. Without it, the

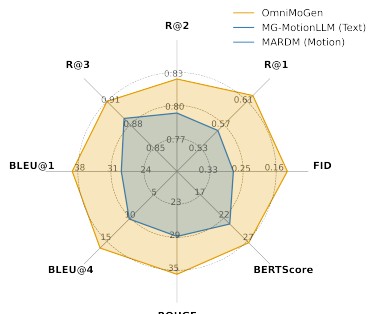

Figure 2: Comparisons of future-aware behavior understanding results.

Table 5: The effectiveness of the vision modality. We evaluate MoGIC on the motion generation and in-between tasks conditioning on different conditions without finetuning. *L, V, M* represent language, vision, and motion, respectively.

| Category | Task | R Precision↑ | | | FID↓ | Matching↓ | Diversity↑ |
|----------|------|------|------|------|------|-----------|-----------|
| | | Top 1 | Top 2 | Top 3 | | | |
| Motion Gen. | *L2M* | $0.590^{\pm0.002}$ | $0.804^{\pm0.001}$ | $0.897^{\pm0.002}$ | $0.330^{\pm0.003}$ | $1.913^{\pm0.007}$ | $12.757^{\pm0.075}$ |
| | *V2M* | $0.408^{\pm0.002}$ | $0.639^{\pm0.003}$ | $0.789^{\pm0.003}$ | $0.634^{\pm0.010}$ | $2.881^{\pm0.012}$ | $12.662^{\pm0.108}$ |
| | *LV2M* | $0.589^{\pm0.001}$ | $0.801^{\pm0.001}$ | $0.898^{\pm0.002}$ | $0.266^{\pm0.002}$ | $1.953^{\pm0.007}$ | $12.585^{\pm0.067}$ |
| Motion In-Bet. | prefix | $0.498^{\pm0.002}$ | $0.720^{\pm0.002}$ | $0.830^{\pm0.001}$ | $0.436^{\pm0.004}$ | $2.373^{\pm0.006}$ | $12.469^{\pm0.052}$ |
| | prefix *w/ L* | $0.624^{\pm0.001}$ | $0.833^{\pm0.001}$ | $0.918^{\pm0.001}$ | $0.137^{\pm0.001}$ | $1.707^{\pm0.003}$ | $12.487^{\pm0.041}$ |
| | prefix *w/ V* | $0.553^{\pm0.001}$ | $0.766^{\pm0.001}$ | $0.868^{\pm0.001}$ | $0.205^{\pm0.001}$ | $2.021^{\pm0.004}$ | $12.639^{\pm0.058}$ |
| | prefix *w/ L+V* | $0.619^{\pm0.001}$ | $0.830^{\pm0.001}$ | $0.914^{\pm0.001}$ | $0.132^{\pm0.001}$ | $1.701^{\pm0.004}$ | $12.662^{\pm0.065}$ |

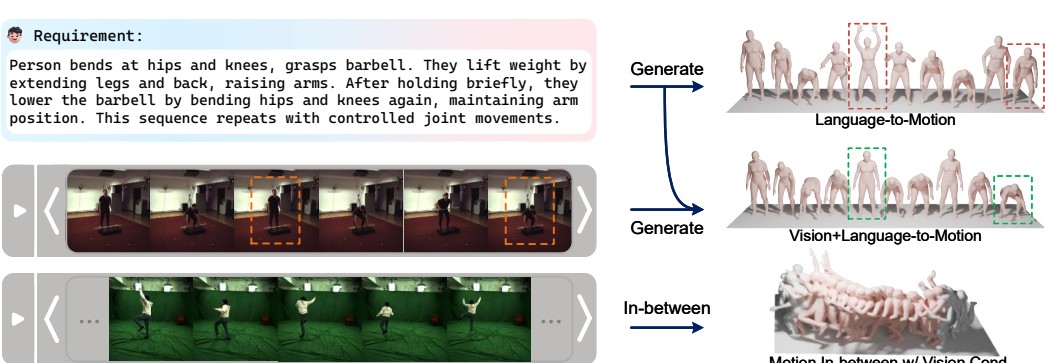

Figure 3: Visualization of motion generation and motion in-between tasks with vision modality.

model relies on shallow correlations, failing to capture the causal structure and latent goals of human motion. Consequently, generated motions lose realism.

**Effectiveness of Training with Vision Modality** We assess the contribution of vision from two complementary angles. (i) *Vision as priors.* We drop vision-to-motion and vision-language-to-motion losses in the cross-modal generative training (MoGIC *L2M* + *Und. loss* in Table 2). Compared to MoGIC *w/ L2M FT* which is finetuned on language-to-motion and future-aware behavior understanding losses after the complete cross-modal generative training, training without vision modality leads to degraded language-to-motion performance, indicating that the visual modality enables the model to learn richer contextual representations and implicitly guides the alignment between generated motions and their conditioning inputs. (ii) *Vision as a conditioning modality.* We further examine whether adding vision conditions improves generation. On Mo440H-MLV, we evaluate both vision–language-to-motion and vision-based motion in-between without task-specific fine-tuning. As shown in

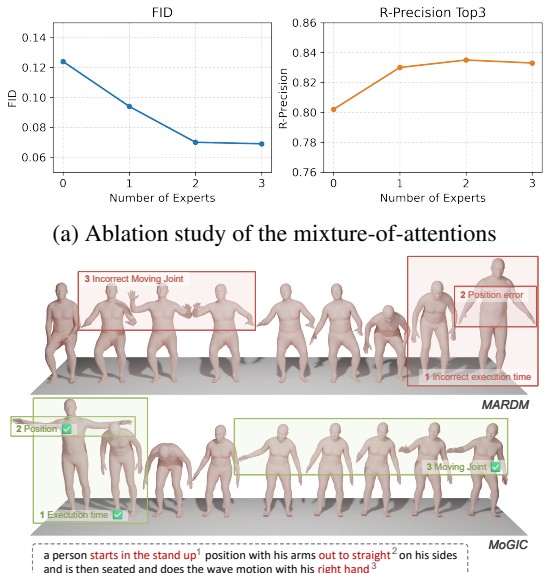

(a) Ablation study of the mixture-of-attentions

(b) Visualization results of L2M.

Figure 4: The effectiveness of mixture-of-attention.

Table 5, vision consistently reduces FID while keeping diversity comparable, and combining language with vision yields the best trade-off. This suggests that visual conditions provide complementary spatiotemporal hints beyond text or motion alone, leading to more natural and coherent generations.

**Effectiveness of Mixture-of-Attention**   We test four settings: no expert (i.e., no cross-attention with fine-grained conditions), one expert ($k_1 \in [0, \infty)$), two experts ($k_1 \in [0, \infty), k_2 \in [1, 6)$), and three experts ($k_1 \in [0, \infty), k_2 \in [1, 6), k_3 \in [6, 10)$). As shown in Figure 4a, fine-grained conditions greatly boost retrieval performance. Increasing expert number steadily reduces FID, with retrieval precision peaking at two experts. To balance efficiency and effectiveness, we adopt two experts as default. Figure 4b further shows that, thanks to mixture-of-attention with adaptive scope, our method generates motions with more precise local responses, including joint movement, positioning, and timing.

**Impact of Video Sampling Strategy**   We further analyze how different visual sampling strategies affect motion generation quality. Since the visual stream in MoGIC is designed to provide weak conditions rather than dense kinematic supervision, we compare (1) sparse 1 FPS inputs (default setting), (2) denser 2 FPS inputs,

Table 6: Effect of video sampling strategies on IDEA400.

| Sampling | FID↓ | Top1↑ | Top2↑ | Top3↑ | Div.↑ | Match↓ |
|---|---|---|---|---|---|---|
| 1 FPS | 0.443 | 0.285 | 0.466 | 0.588 | **9.106** | 3.321 |
| 2 FPS | 0.423 | 0.284 | **0.469** | **0.597** | 8.903 | **3.226** |
| Flow-based | 0.421 | **0.286** | 0.468 | 0.591 | 8.797 | 3.304 |
| Acc.-based | **0.418** | 0.285 | 0.468 | 0.593 | 8.817 | 3.302 |

and (3) dynamic-aware keyframes selected by optical-flow magnitude and joint-acceleration peaks (10 frames). As shown in Table 6, while denser or motion-centric sampling slightly improves FID, it consistently reduces motion diversity, indicating that the model becomes over-constrained by deterministic visual poses and tends toward keyframe interpolation. In contrast, 1 FPS inputs already capture essential context, interaction type, and coarse movement tendencies, yielding a strong balance between fidelity and generative flexibility.

**The proportion of visible motion prefixes in Future-Aware Behavior Understanding**   We investigate how the proportion of visible motion prefixes used in the future-aware behavior understanding task influences downstream motion generation. As shown in Table 7, providing a moderate amount of prefix motion (50%) leads to the best overall performance,

Table 7: Ablation study of future-aware behavior understanding.

| Visible Rate | Top1↑ | Top2↑ | Top3↑ | FID↓ | Match↓ | Div.↑ |
|---|---|---|---|---|---|---|
| 100% | 0.646 | 0.845 | 0.919 | 0.198 | 1.910 | **12.623** |
| 75% | 0.648 | 0.847 | 0.922 | 0.171 | 1.905 | 12.599 |
| 50% | **0.652** | **0.851** | **0.926** | **0.134** | **1.889** | 12.434 |
| 25% | 0.650 | 0.848 | 0.923 | 0.156 | 1.905 | 12.417 |

as it offers sufficient contextual cues while still encouraging the model to infer future dynamics. In contrast, either exposing the entire prefix or revealing too little motion reduces generation quality, indicating that balanced prefix visibility is essential for effective future-aware behavior understanding.

## 6   CONCLUSION

In this work, we introduce MoGIC, a unified multimodal framework that couples future-aware behavior understanding with multimodal-conditioned motion generation. By jointly modeling high-level future semantic patterns and continuous motion synthesis across language, vision, and motion, MoGIC effectively resolves ambiguities inherent in text-only conditioning and delivers versatile generative capability. To support this paradigm, we construct Mo440H, a 440-hour tri-modal benchmark aggregated from 21 diverse motion datasets. Comprehensive experiments across HumanML3D and Mo440H verify MoGIC's substantial gains in generation fidelity, captioning performance, and multi-conditioned synthesis, including vision-guided generation and future-behavior understanding. We believe these findings offer new insights into multimodal human motion understanding and lay the groundwork for more precise, and semantically grounded motion generation in future research.

ETHICS STATEMENT

**Dataset usage and compliance.**   All datasets used in this work are publicly available and can be freely downloaded from the internet. We strictly comply with the official licenses and usage terms associated with each dataset, ensuring that our data use aligns with community norms and legal requirements.

**Privacy and anonymity.**   The datasets employed do not contain personally identifiable information (PII) or sensitive private data. Our work involves only motion capture–style representations (e.g., 3D skeleton joints, body parameters), which are inherently abstract and anonymized.

**Human subjects.**   No human subject experiments were conducted for this research. All experimental results are derived from existing open-source datasets, which have already been collected and released by their respective authors. Thus, issues related to IRB approval, informed consent, or direct participant involvement are not applicable.

**Application scope and potential risks.**   This research falls under the category of generative modeling, with primary applications in animation, computer graphics, and artistic content creation. We do not foresee potential risks of harmful applications, such as surveillance, discrimination, or misuse in security-critical settings.

**Representation and likeness.**   The generated outputs represent human body kinematics in terms of parameters such as joint positions and motion trajectories. They do not reproduce personal likenesses, facial identities, or biometric information, and therefore do not raise concerns about portrait rights or identity misuse.

**Fairness, bias, and integrity.**   The datasets used are diverse motion corpora but, as with all public datasets, may contain imbalances in action types or distributions. Our focus is on methodological contributions rather than demographic or identity-sensitive attributes. We confirm that all results presented in this paper are genuine, and no data manipulation or misrepresentation has been performed.

REPRODUCIBILITY STATEMENT

**Model description.**   The main body of the paper provides a comprehensive description of our proposed architecture, including the conditional masked transformer, diffusion head, and the disentangled generation modules.

**Training and evaluation details.**   Hyperparameters, optimization strategies, and hardware settings are reported in the experimental section. We also describe the evaluation metrics, number of epochs, and sampling steps to ensure clarity and transparency.

**Dataset processing.**   The composition of the datasets used in the experiments is reported in the main text. Further details on sequence segmentation, normalization, and preprocessing are provided in the supplementary materials.

**Code availability.**   An anonymous link to the source code is provided in the supplementary materials. The repository contains scripts for training, evaluation, and dataset preparation, allowing other researchers to replicate our results.

**Documentation of assumptions.**   All assumptions made in model design and implementation are explicitly documented in the paper and supplementary materials, enabling verification and reproducibility of our findings.

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
