

Figure A1: Visualization of data distributions. For each dataset, we randomly sample 2,000 motion sequences. Each sequence is temporally averaged to obtain a compact feature representation, which is then reduced in dimensionality using t-SNE.

# A APPENDIX

## A.1 MO440H MOTION DATASET

Some existing approaches estimate human motion directly from web videos. However, such video data often suffer from motion degradation caused by low image quality, occlusions, and frequent viewpoint changes, which result in unstable monocular pose estimation and ultimately compromise the quality of motion generation at the data level. In addition, these datasets typically exhibit limited motion diversity. For instance, many MotionX (Lin et al., 2023) samples come from repetitive video series (e.g., yoga tutorials), while MotionMillion (Fan et al., 2025) is dominated by sports-related activities such as martial arts, fitness, and dance.

To ensure both data quality and coverage across diverse motion scenarios, we construct Mo440H, a unified benchmark built from 21 high-quality open-source motion datasets. We standardize properties such as frame rate, motion orientation, and maximum sequence length, while integrating motions spanning single-person activities, human–object interactions, and multi-person interactions. The included datasets are as follows. Furthermore, we visualize the dataset distribution in Figure A1. For each dataset, we randomly sample 2,000 motion sequences, temporally average each sequence into a compact feature representation, and then apply t-SNE for dimensionality reduction.

*BABEL (Punnakkal et al., 2021)* A large-scale motion-language dataset ( 43 h) from AMASS, providing 28k labeled sequences and 63k frame-level annotations across 250 categories for action recognition, segmentation, and motion-language understanding.

*BEAT (v2) (Liu et al., 2024a)* A 76 h multi-modal co-speech gesture dataset with 30 speakers in 4 languages, offering synchronized body motion, facial expression, audio, text, and emotion for gesture generation research.

*Fit3D (AIFit) (Fieraru et al., 2021)* A fitness motion dataset with 13 subjects performing 37 exercises, containing 3 M RGB frames aligned with 3D poses, supporting pose estimation and automated training feedback.

*HuMMan (Cai et al., 2022)* A massive multi-modal 4D dataset of 1000 subjects and 500 actions ( 60 M frames), providing synchronized RGB, depth, IMU, skeletons, SMPL, and textured meshes for comprehensive human modeling.

*HumanML3D (Guo et al., 2022)* A motion-language dataset with 14,616 motion clips paired with 44,970 captions, enabling text-conditioned motion generation and motion-to-language tasks.

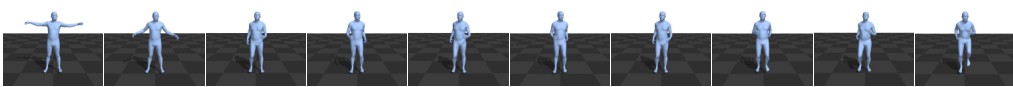

**VLM output**: A human figure stands with arms outstretched, then lowers them to the sides. The figure begins jogging in place, arms pumping rhythmically. Legs alternate lifting off the ground, simulating a running motion. The body remains upright throughout the sequence.

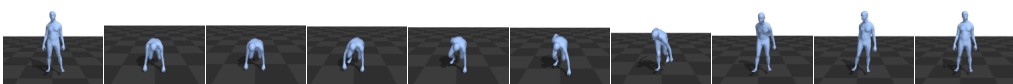

**VLM output**: A figure starts in a crouched position, hands and feet on the ground. It then pushes off with its hands, extending its legs and rising to a standing position. The figure stands upright with arms relaxed at its sides, completing the motion smoothly and fluidly.

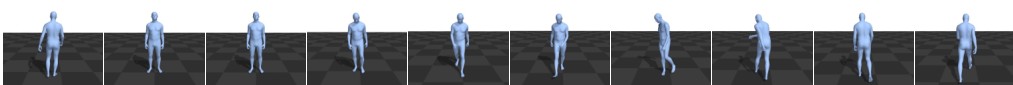

**VLM output**: A blue human figure stands still, then begins walking forward. The figure moves its arms naturally with each step, transitioning from a frontal view to a rear view as it walks away. The motion is smooth and continuous, showcasing a typical walking gait.

Figure A2: Some examples of motion descriptions generated by Qwen2.5-VL-Max (Bai et al., 2025). Although the input is a human mesh sequence rendered based on SMPL parameters, the Qwen model still outputs a relatively accurate description.

*IDEA-400 (Lin et al., 2023)* A dataset of 13k sequences across 400 action categories with text descriptions, offering a benchmark for generalizable motion generation and recognition.

*MoYo (Tripathi et al., 2023)* A yoga-focused dataset with 200 poses (1.75 M frames) captured by multi-camera and pressure-sensing mat, providing SMPL-X fits and contact data for studying balance and extreme poses.

*MPI-INF-3DHP (Mehta et al., 2017)* A 3D pose dataset of diverse single-person activities in indoor/outdoor settings, captured markerlessly with multi-cameras for in-the-wild pose estimation benchmarks.

*MVHumanNet (Xiong et al., 2024)* A large-scale multi-view dataset with 4,500 identities and 60k sequences (645 M images), annotated with segmentation and calibration for digital human reconstruction under clothing variations.

*Motion-X (Lin et al., 2023)* A 3D whole-body dataset of 96k sequences (13.7 M frames) in SMPL-X format, capturing expressive motions with hand and face details, with text labels for ¿80k sequences. Although this dataset is derived from web data, we still include it in Mo440H to enhance diversity.

*EgoBody (Zhang et al., 2022)* An egocentric multi-person dataset with 125 sequences, combining HoloLens2 RGB-D, gaze, and ground-truth 3D meshes for first-person interaction understanding.

*InterHuman (InterGen) (Liang et al., 2024b)* A large-scale two-person interaction dataset ( 107 M frames) with 16,756 textual descriptions, covering diverse social and cooperative motions.

*HOI-M³ (Zhang et al., 2024a)* A multi-person multi-object dataset ( 181 M frames) of 46 subjects and 90 objects, recorded by 42 cameras and IMUs for studying complex group interactions.

*CORE4D (Liu et al., 2024b)* A dataset of 1,000 real-world and 10k+ augmented collaborative rearrangement sequences, focusing on multi-human multi-object cooperation in household scenes.

*CIRCLE (Araújo et al., 2023)* A contextual dataset of 10 h reaching/interaction motions across 9 scenes with synchronized egocentric RGB-D, enabling study of human–scene relations.

*CHAIRS (Jiang et al., 2023b)* A dataset of 17.3 h interactions between 46 people and 81 articulated chairs, with aligned human–object meshes for studying posture and manipulation.

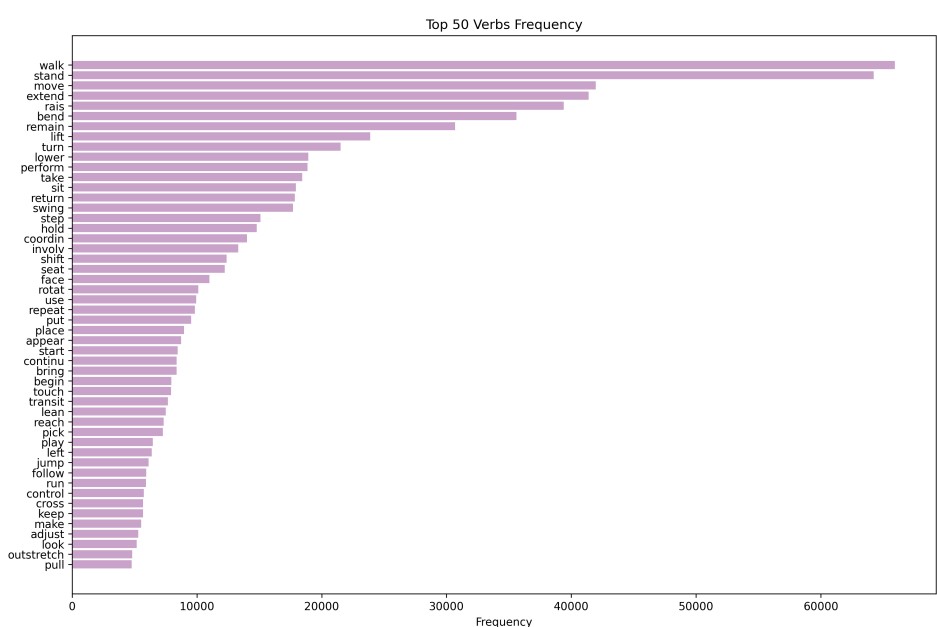

Figure A3: Verb frequency of text annotations in our integrated Mo440H.

*GRAB (Taheri et al., 2020)* A whole-body grasp dataset with 10 subjects interacting with 51 objects, providing detailed body, hand, face, and object contact data for realistic manipulation modeling.

*HIMO (Lv et al., 2025)* A benchmark with 3,300 sequences (4.08 M frames) of long, multi-object interactions paired with fine-grained text annotations for complex activity understanding.

*HUMANISE (Wang et al., 2022)* A synthetic dataset aligning mocap motions with 3D indoor scenes, annotated with language descriptions to enable motion generation conditioned on scene context.

*IMHD$^2$ (Zhao et al., 2024)* A high-dynamic interaction dataset (295 sequences, 892k frames) captured with high-speed cameras and IMUs, featuring fast-motion tasks with ground-truth body and object trajectories.

*OMOMO (Li et al., 2023)* A conditional dataset of 10 h sequences coupling object trajectories with human reactions, providing human–object pairs for learning motion conditioned on object dynamics.

**Data Annotation** For datasets that contain visual modality but lack textual annotations, we first segment motion sequences with a maximum length of 10 seconds and then use Qwen2.5-VL-Max to automatically generate descriptions for each sequence. The annotation prompt is as follows:

*Briefly describe the human motion, focusing on the interaction with objects and body movements. Do not describe text that appears in the video. Describe objectively. The output must be in one paragraph and no more than 100 words. Do not describe the time explicitly.*

Some datasets do not provide RGB video modality; instead, they render virtual motion videos from SMPL body representations. For these datasets, we also employ Qwen to perform annotations. We observe that Qwen demonstrates strong robustness in handling such virtual humans and produces reasonably accurate motion descriptions. Figure A2 illustrates several annotation examples on rendered virtual humans.

To ensure annotation quality, we apply a three-stage filtering pipeline to all VLM-generated labels. First, we perform dataset-level rejection by sampling captions from each dataset and discarding datasets whose visual modality does not reliable. For instance, Chairs is excluded because its extremely sparse visual frames prevent the model from capturing temporal continuity. Second, we conduct model-based rejection by training an early version of our motion generator on the auto-

Table A1: Human evaluation comparing human-written and Qwen-generated captions. Raters performed blind A/B assessment on 200 randomly sampled sequences.

| Source | Description accuracy↑ | Linguistic coherence↑ | Detail richness↑ |
|---|---|---|---|
| Human-annotated | **4.05** | **4.10** | 3.58 |
| Qwen-annotated | 3.84 | 4.06 | **4.18** |

Table A2: Zero-shot evaluation on the human-labeled HuMMan dataset. Adding Qwen-generated captions significantly improves generalization.

| Train dataset | FID↓ | Top1↑ | Top2↑ | Top3↑ | Diversity↑ | Matching↓ |
|---|---|---|---|---|---|---|
| HumanML3D+HIMO+InterHuman | 29.912 | 0.046 | 0.090 | 0.122 | 3.868 | 9.958 |
| All w/o HuMMan | **13.544** | **0.077** | **0.150** | **0.207** | **6.422** | **7.532** |

captioned data and identifying datasets that degrade generation performance. In this stage, although PROX captions are broadly correct, heavy occlusions cause ambiguous fine-grained motion and thus its annotations are removed. Finally, we manually proofread randomly selected captions from the remaining datasets. Since all motions originate from clean mocap recordings, we observe a low error rate after this stage. We show the frequency of verbs in the text annotations in Figure A3.

**Motion Representation**   To unify motion representations across heterogeneous datasets, we convert all sequences into global joint positions via SMPL/SMPL-X fitting and extract a consistent 22-joint skeleton following HumanML3D and MARDM conventions. We resample all motions to 30 fps, align characters to face the positive $z$-axis in the first frame, trim sequences longer than 10 seconds, and discard those shorter than 2 seconds. Based on MARDM, we compute root angular velocity, root linear velocity on the ground plane, root height velocity, and local joint positions, producing a standardized 67-dimensional representation.

**Human Evaluation of Qwen-Generated Labels**   To further validate the reliability of Qwen-generated annotations, we conducted a controlled human evaluation. We randomly sampled 200 Mo440H sequences, manually re-annotated them, and collected blind A/B ratings from ten independent evaluators. Raters compared human-written and Qwen-written captions without knowing their origin and scored each on three criteria: description accuracy, linguistic coherence, and detail richness. As shown in Table A1, Qwen-generated captions achieve accuracy and fluency on par with human-written text, while providing significantly richer motion details. These results confirm that Qwen captions constitute high-quality supervisory signals suitable for large-scale motion generation training.

**Zero-Shot Validation on a Purely Human-Labeled Dataset**   We further examine whether training on Qwen-generated labels introduces undesirable bias that harms generalization. Two models are trained: (i) a human-only model using HumanML3D, HIMO, and InterHuman, and (ii) a model trained on the full dataset with both human and Qwen-generated labels, excluding HuMMan in both cases. Zero-shot evaluation is conducted on HuMMan, which is entirely human-annotated and unseen during training. As shown in Table A2, incorporating Qwen-generated labels substantially improves zero-shot generalization, reducing FID from 29.91 to 13.54 and consistently boosting Top-k accuracy, diversity, and matching scores. This demonstrates that Qwen-generated labels provide effective and unbiased supervision that enhances true generalization beyond human-only training.

## A.2 EVALUATION METRICS

For motion generation, we use several common quantitative metrics to evaluate different aspects of model performance. FID (Fréchet Inception Distance) measures how close the distribution of generated motions is to real motions, reflecting overall realism. R-Precision and Matching Score evaluate whether the generated motions match the given text descriptions, focusing on text–motion alignment. To check the diversity of outputs, we report Multimodality Diversity, which measures how much variation the model can produce under the same condition. We also include CLIP-Score

Table A3: Zero-shot evaluation on the unseen IDEA400 dataset. MoGIC consistently outperforms MARDM under both HumanML3D-only and Mo440H (excluding IDEA400) training settings.

| Method | Train data | FID↓ | Top1↑ | Top2↑ | Top3↑ | Diversity↑ | Matching↓ |
|--------|-----------|------|-------|-------|-------|-----------|-----------|
| MARDM | HumanML3D | 13.968 | 0.134 | 0.215 | 0.328 | 7.654 | 5.669 |
| MoGIC | HumanML3D | **9.912** | **0.156** | **0.248** | **0.360** | **7.885** | **5.369** |
| MARDM | Mo440H-ML w/o IDEA400 | 6.344 | 0.228 | 0.374 | 0.502 | 10.221 | 3.467 |
| MoGIC | Mo440H-ML w/o IDEA400 | 4.269 | 0.242 | 0.403 | 0.541 | 10.747 | 3.349 |

Table A4: Zero-shot compositionality evaluation on FineMotion's re-annotations of HumanML3D. MoGIC achieves better alignment with fine-grained textual descriptions.

| Method | FID↓ | Top1↑ | Top2↑ | Top3↑ | Diversity↑ | Matching↓ | CLIP score↑ |
|--------|------|-------|-------|-------|-----------|-----------|-------------|
| MARDM | 35.688 | 0.075 | 0.141 | 0.192 | 7.001 | 6.240 | 0.402 |
| MoGIC | **30.740** | **0.087** | **0.157** | **0.220** | **6.985** | **5.170** | **0.497** |

(following Meng et al. (2025)), which leverages pretrained vision–language models to further test semantic consistency between motion and text.

For text generation, we evaluate the captions generated for motions using both n-gram and semantic metrics. BLEU@4 and BLEU@1 measure overlap at different n-gram levels, while ROUGE emphasizes recall of important phrases. To capture similarity beyond exact wording, we report BERTScore, which uses contextual embeddings to measure semantic closeness to reference captions.

## A.3 VISUALIZATION RESULTS

**Visualization of Language-to-Motion** Figure A4 presents a comparison of language-to-motion training and testing results on the HumanML3D dataset. We visualize MoGIC alongside MARDM and MotionAgent. As shown, motions generated by MoGIC exhibit more coherent behavioral logic and higher realism. Figure A5 further provides zero-shot inference examples after training on Mo440H, where MoGIC performs particularly well on tasks with causal dependencies, such as picking up and placing objects.

**Visualization of Vision+Language-to-Motion** Figure A6 compares vision+language-to-motion generation with language-only generation under various interaction scenarios. In the chair interaction task, the language-only setting understands the intended action sequence ("move the chair, sit down, shift weight") but lack information about the chair's actual position, resulting in misaligned motions. In contrast, vision and language-conditioned MoGIC receives the precise chair location and human–chair geometry from video frames, enabling accurate contact, and sitting behaviors.

For the dumbbell-raising task, textual descriptions such as "raise dumbbells overhead" provide only coarse description, leading to inaccurate movements. With visual input, MoGIC captures fine-grained kinematic details, such as synchronized shoulder-elbow motion, subtle knee flexion during lowering, and realistic pose limits, resulting in more faithful upper-body dynamics.

We further include examples where rendered human meshes and object meshes are used as visual inputs. We can ses that visual frames provide the exact box location, hand-object contact patterns, and movement trajectory. MoGIC consequently produces coherent lifting, carrying, and placing behaviors, whereas language-only models often hallucinate contact or generate physically implausible motions.

## A.4 ZERO-SHOT ABILITY

To assess MoGIC's robustness under out-of-distribution conditions, we conduct two dedicated zero-shot studies. First, in Table A3 we evaluate generalization to an unseen dataset, IDEA400, by training both MoGIC and MARDM on either HumanML3D or Mo440H (excluding IDEA400). MoGIC consistently outperforms MARDM across all settings, demonstrating stronger transfer of motion se-

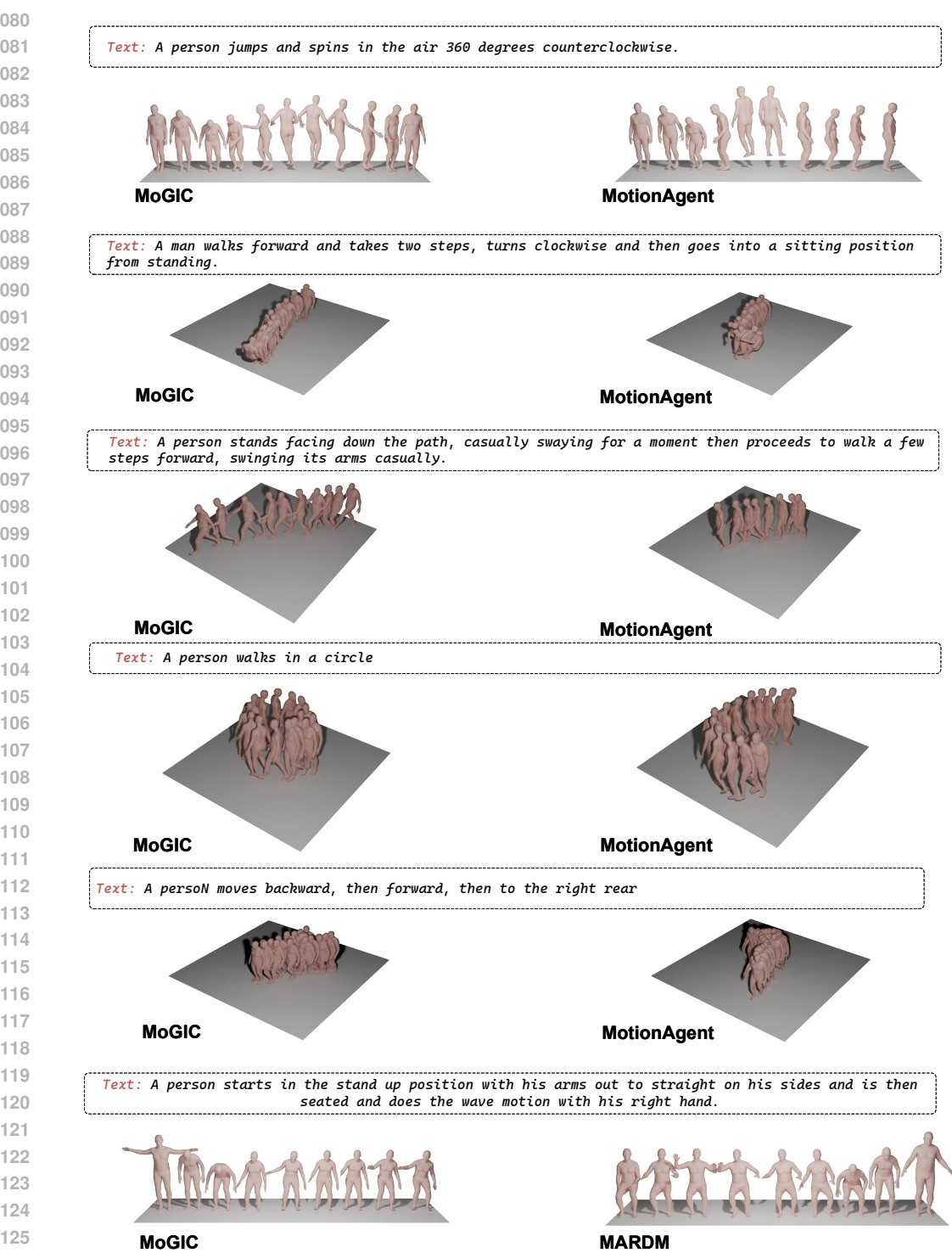

Figure A4: Visualization results of language-to-motion generation.

mantics beyond the training distribution. Second, we examine compositional generalization under unseen textual styles using the FineMotion re-annotations of HumanML3D, where motions remain identical but descriptions are substantially more fine-grained (Table A4). Despite never observing this annotation style during training, MoGIC yields better FID, higher CLIP score and R-precision, and lower Matching Score compared to MARDM. However, the absolute performance remains lim-

*Text: A person transitions from a crouched position to standing upright. They extend their legs and straighten their back, raising their arms outward. The individual maintains balance with feet apart, arms outstretched, and head tilted back, showcasing flexibility and control in their movements.*

*Text: A person picks up the box from the ground, walks forward, and then puts the box down.*

*Text: A person pushes the object in front away, then sits down.*

*Text: A person bends down to pick up a box, holds it against the chest, walks a few steps forward, and then carefully places the box onto a table.*

*Text: A person squats down, places both hands on their knees, then stands up and walks forward.*

*Text: A person who is running.*

*Text: A person picks up the object from the ground and hand it to the person in front.*

Figure A5: Visualization results of language-to-motion generation.

ited due to the substantial stylistic gap between HumanML3D and FineMotion annotations, the fine-grained nature of the latter, and the mismatch between these detailed descriptions and the evaluator.

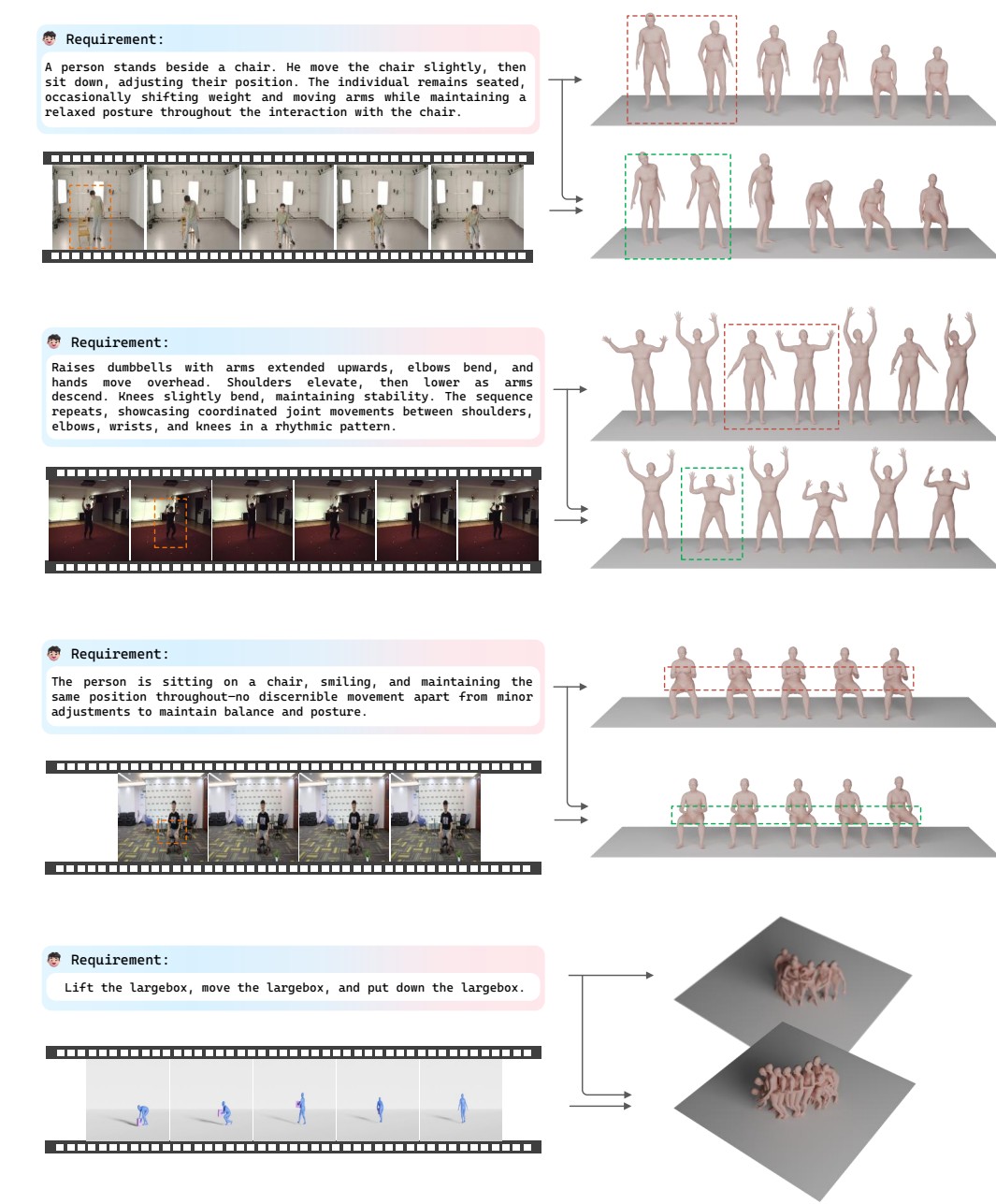

Figure A6: Visualization results of vision+language-to-motion generation.

## A.5 INFERENCE SPEED

All inference experiments are conducted on a workstation equipped with an AMD 9950X CPU and an NVIDIA RTX 5090 GPU. We measure inference efficiency as the wall-clock latency for generating 32 samples in parallel. We analyze two major factors that affect inference speed and performance:

**Number of Iterations.** Our model consists of a conditional masked transformer and a diffusion head and the transformer is followed the training pipeline of masked modeling (Meng et al., 2025; Pinyoanuntapong et al., 2024b). Inference starts from a fully masked latent variable, and the model reconstructs the masked regions iteratively. We set the default number of iterations to 17, and further

Table A5: Ablation study of iteration number in masked modeling and sampling steps in diffusion.

| Method | Sampling Steps | Iter. Number | R Precision↑ | | | FID↓ | Match↓ | Infer. Time(s) |
|---|---|---|---|---|---|---|---|---|
| | | | Top 1 | Top 2 | Top 3 | | | |
| MoGIC | 20 | 17 | 0.540 | 0.740 | 0.832 | 0.079 | 3.02 | 1.883 |
| | 10 | 17 | 0.545 | 0.741 | 0.835 | 0.070 | 3.00 | 0.840 |
| | 8 | 17 | 0.530 | 0.735 | 0.829 | 0.071 | 3.02 | 0.683 |
| | 5 | 17 | 0.525 | 0.733 | 0.826 | 0.068 | 3.04 | 0.418 |
| | 2 | 17 | 0.509 | 0.705 | 0.813 | 0.378 | 3.15 | 0.193 |
| | 10 | 5 | 0.545 | 0.742 | 0.840 | 0.138 | 2.98 | 0.255 |
| | 10 | 10 | 0.534 | 0.742 | 0.835 | 0.079 | 3.01 | 0.493 |
| | 10 | 25 | 0.537 | 0.734 | 0.837 | 0.073 | 3.01 | 1.095 |
| MARDM | - | 17 | 0.500 | 0.695 | 0.795 | 0.114 | 3.270 | 9.813 |
| MMM | - | 10 | 0.487 | 0.683 | 0.782 | 0.132 | 3.359 | 0.358 |

vary this value to examine its effect on inference speed and performance. Results are reported in Table A5.

**Number of Diffusion Sampling Steps.** The diffusion module adopts Euler sampling with 10 default steps. Unlike MARDM (Meng et al., 2025), which relies on adaptive solvers (dopri5), our approach generates realistic motions with only a small number of steps. We also compare different numbers of sampling steps and report the corresponding runtime and performance in Table A5.

These experiments demonstrate that, compared with other diffusion-based methods, our model achieves a clear advantage in real-time inference, maintaining competitive performance even with very few sampling steps.

## LLM USAGE STATEMENT

We made limited use of large language models (LLMs) during the preparation of this work, and we accept full responsibility for all content presented in the paper.

**Writing support.** LLMs (Qwen2.5 and Qwen3) were used to suggest alternative phrasings and to refine grammar in early drafts of the abstract, introduction, and related work sections. All technical descriptions, theoretical claims, and citations were written and verified solely by the authors.

**Research support.** LLMs were occasionally employed for ideation, such as exploring possible ablation settings and generating generic utility code (e.g., configuration scripts and logging utilities). All experimental code, implementations of the proposed method, and reported results were developed, validated, and confirmed by the authors.

**Data and annotation.** For automated text annotations, we employed the LLM Qwen2.5-VL. The details of this process are described in the paper, and the exact prompts and cached responses are included in the anonymous supplementary materials. Outputs were manually reviewed and corrected where necessary before use.

No LLM has been credited as an author. All outputs were checked for factual accuracy, correctness, and relevance, and the authors remain fully accountable for the entirety of this work.