# OpenReview forum: "MoGIC: Boosting Motion Generation via Intention Understanding and Visual Context"
_ICLR.cc/2026/Conference — Submitted to ICLR 2026_

### Official Review · Reviewer_1tfi · 2025-10-21

**Soundness:** 1
**Presentation:** 2
**Contribution:** 2
**Rating:** 4
**Confidence:** 4

**Summary:**

MoGIC proposes a multimodal framework for human motion generation that fuses text, (optionally) vision, and partial motion via a Conditional Masked Transformer (CMT). The CMT applies (i) global semantic-level modulation (adaptive LayerNorm) and (ii) a mixture-of-attention with adaptive Top-k cross-attention to select relevant text/vision snippets. A diffusion-style Motion Generation Head (MGH) produces motions; a lightweight T5-style Intention Prediction Head (IPH) generates an explicit textual “intention.”

They train across five tasks (L2M, VL2M, V2M, M2M, and intention prediction) with a unified loss, truncating the last 50% of motion tokens for IPH.  The integrated dataset “Mo440H” (≈440 hours, 21 sources) is constructed; for datasets without captions they use Qwen2.5-VL to auto-caption, and for datasets lacking RGB they render mesh sequences to videos, downsampled to 1 fps. They also evaluate on HumanML3D, using prior evaluators there and a retrained evaluator for Mo440H.

**Strengths:**

* A coherent architecture that disentangles continuous motion generation (diffusion/interpolant) from discrete intention text, while injecting multimodal context at both semantic and token levels. The block-level design is sensible and easy to slot into existing masked-token pipelines.
* Cross-modal joint training across L2M / V2M / M2M / IP is a clean way to amortize supervision and can plausibly improve controllability/faithfulness.

**Weaknesses:**

1. The dataset construction lacks details. Auto-captioning with Qwen2.5-VL and manual filtering is fine, but release the prompts, filtering criteria, and failure cases. Likewise, clearly mark which entries use rendered meshes vs real RGB, and how those are used in each task. Different datasets used different motion representation. What representation does Mo440H use? How did the authors convert different joint representation?

2. For sources without RGB, the “vision” is rendered from the motion itself (“rendered mesh sequences were adopted instead,” 1 fps). If these rendered frames are used in training/evaluation for V2M or VL2M, the model is being conditioned on images deterministically derived from the target motion—i.e., target leakage across modalities. This can inflate the apparent benefit of visual conditioning and overstate real-world applicability (where visuals are independent RGB videos). The paper should explicitly exclude such cases from V2M/VL2M evaluation or report separate results using only genuine RGB.


3. “Intention” is underspecified and may collapse to partial-captioning.
The task defines intention as text generated from the first 50% of a motion and calls it a “conceptual goal.” In practice, this looks like captioning of observed partial motion rather than predicting latent goals (which would require causal/goal semantics, not just n-gram overlap). The metrics shown (BLEU/ROUGE/BERTScore) mostly measure textual similarity, not intention correctness.

4. The authors did not specify the train/eval split on the new dataset. They retrain an evaluator for Mo440H “following previous methods.” Without clear held-out splits to reproduce the evaluator, cross-paper comparability and data leakage risks remain. For T2M evaluation on HumanML3D, did the authors train the model on whole Mo440H or HumanML3D alone?

**Questions:**

1. See **Weakness 1**.

2. Please justify or explain **Weakness 2**.

3. For intention task: What is the ground truth? If it’s an LLM caption of the observed prefix, how do you ensure it reflects “intent,” not mere surface description? Any human study or goal-specific metrics?

4. See **Weakness 4**.

---

> ### Author Response · Authors · 2025-11-25
> **Response to Reviewer 1tfi (1/3)**
>
> We sincerely thank the reviewers for their constructive feedback. We appreciate the time and effort invested in reading our manuscript and providing valuable suggestions. We have carefully addressed comments below.
>
> > **W1: Details of dataset construction, prompts, filtering criteria, and motion representation**
>
> **(a) Prompt for auto-captioning**
>
> For all datasets, we use the same prompt shown in Appendix A.1.
>
> **(b) Filtering criteria for VLM-generated labels**
>
> Our filtering pipeline included three stages:
>
> 1. **Dataset-level rejection.**
> For each dataset, we sampled captions and discarded datasets whose visual modality cannot support reliable captioning. In this stage, Chairs dataset was rejected because its sparse frames cause Qwen2.5-VL to miss motion continuity.
>
> 2. **Model-based rejection**
> We trained an early version of the motion generator on auto-captioned data and examined which datasets degrade generation quality. In this stage, the text annotations of PROX was rejected because occlusions make fine-grained motion ambiguous, even though captions were broadly correct.
>
> 3. **Manual proofreading on remaining datasets.**
> We manually verified and corrected a randomly sampled subset of captions. Given that the underlying data comes from clean mocap recordings, the resulting error rate was found to be low.
>
> **(c) Marking datasets using rendered meshes vs RGB**
>
> We will clearly annotate this in the revised Appendix:
>
> - Rendered-mesh–based vision: BABEL, GRAB, Humanise, OMOMO
> (GRAB & OMOMO also include object meshes)
> - RGB-based vision: EgoBody, Fit3D, IDEA400, IMHD2, Motion-X, MOYO, MVHumanNet, MPI-INF-3DHP
>
> **(d) Motion representation and conversion**
>
> To handle the heterogeneous joint systems across datasets, we standardize all motions through a unified processing pipeline. We first convert each motion sequence into global joint positions via SMPL/SMPL-X fitting, and then follow the HumanML3D/MARDM conventions to extract a consistent set of 22 joints. Afterward, we normalize every sequence such that the character faces the +z direction in the first frame, sample all motions to 30 fps, trim sequences to ensure a maximum duration of 10 s, and discard sequences shorter than 2 s. Following MARDM, we compute root angular velocity, root linear velocity on the XZ-plane, root height velocity, and local joint positions, yielding a unified 67-dimensional representation. We will fully document this pipeline in the supplementary materials.
>
> > **W2: Leakage concern for V2M/VL2M when vision is rendered from motion**
>
> We thank the reviewer for raising this important concern. We agree that if our task were motion-to-video generation, then using SMPL-rendered frames would indeed cause leakage, because the visual output would be a deterministic rendering of the target motion. However, our setting is the opposite direction: vision/language-to-motion, where the visual modality is intentionally used as a **conditioning signal**. In this formulation, vision, text, and motion are naturally one-to-one aligned modalities describing the same underlying behavior, like how multimodal learning (e.g., CLIP, Video-to-Motion models) routinely pairs images/videos with corresponding motions or captions. This alignment is **not** leakage, but precisely the structure that multimodal training requires.
> Importantly, although the rendered visual sequences are generated from SMPL, their role and information content are no different from ordinary RGB videos: both provide only coarse posture appearance, not the 3D motion representation.
>
> We also report results using only genuine RGB videos and removing all rendered videos during V2M/VL2M:
>
> | **Method** | **FID** | **Top1** | **Top2** | **Top3** | **Diversity** | **Matching** |
> |-----------|---------|----------|----------|----------|---------------|--------------|
> | L2M       | 0.424   | 0.401    | **0.620**    | 0.748    | 10.599        | 2.061        |
> | V2M       | 0.621   | 0.286    | 0.473    | 0.605    | 10.775        | 2.736    |
> | **LV2M**  | **0.363** | **0.405** | 0.618 | **0.750** | **10.653** | **2.056** |
>
> These results confirm that the benefits of visual conditioning persist even when relying solely on genuine RGB videos.
>
> Moreover, in our framework the visual modality is not a strong supervision signal.
>
> - The video stream is downsampled to 1 fps and is temporally unaligned with the motion sequence (30 fps).
> - Visual tokens are injected only through soft semantic cues (global modulation + adaptive Top-k cross-attention), not through regression or reconstruction losses.
>
> Thus the visual modality acts as a weak prior or conditions, rather than transferring deterministically encoded motion.

---

> > ### Author Response · Authors · 2025-11-25
> > **Response to Reviewer 1tfi (2/3)**
> >
> > > **W3 & Q3: Intention may collapse to partial-captioning.**
> >
> > We would like to clarify a key misunderstanding: our Intention Prediction task is not performing partial-captioning of the observed frames, but instead requires predicting a global behavior description that must remain consistent with the unobserved second half of the motion.
> >
> > **(1) Intention Prediction is fundamentally different from partial-captioning**
> >
> > The Intention Prediction task:
> >
> > - removes the entire second half of the motion latent sequence,
> > - requires the model to infer the future-aware behavior semantics that govern both the observed portion and the unobserved future portion,
> > - and outputs a global description, not a summary of the visible frames.
> >
> > Thus, unlike partial-captioning, which describes what has already happened, our task forces the model to learning a structural dependency between early motion and logically consistent global behavior semantics, which we believe is a meaningful form of causal reasoning within the motion domain.
> >
> > **(2) Empirical evidence shows it is not equivalent to caption reconstruction**
> >
> > If Intention Prediction were simply captioning of the visible prefix, then replacing it with a standard captioning loss should yield similar improvements. However, our ablation results show the opposite (Table 2):
> >
> > | Methods                           | Top1            | Top2            | Top3            | FID↓            | Matching↓       | Diversity↑      |
> > |-----------------------------------|-----------------|-----------------|-----------------|-----------------|-----------------|-----------------|
> > | MoGIC *only L2M loss*      | 0.637   | 0.836   | 0.908   | 0.201   | 2.003   | 12.392  |
> > | MoGIC *L2M + Caption loss* | 0.646   | 0.845   | 0.919   | 0.198   | 1.910   | **12.623**  |
> > | MoGIC *L2M + Int. loss*  | **0.652** | **0.851** | **0.926** | **0.134** | **1.889** | 12.434  |
> >
> > When we introduce the Intention Prediction loss (MoGIC *L2M + Int. Loss*), **the FID drops by 33.3%** relative to the baseline (from 0.201 → 0.134). In contrast, adding a captioning loss (MoGIC *L2M + Caption. Loss*), which only reconstructs full-sequence text, yields a negligible 1.5% reduction (from 0.201 → 0.198).
> >
> > A similar trend appears on HumanML3D (Table 1), where removing intention loss leads to the largest degradation across all metrics. This indicates that the benefit is not only from reconstructing captions, but from learning future-aware behavior semantics.
> >
> > **(3) Metrics only evaluate text similarity**
> >
> > While BLEU/ROUGE/BERTScore measure textual similarity, the substantive impact of Intention Prediction appears in motion quality. Across both HumanML3D and Mo440H, incorporating Intention Prediction yields the largest improvements among all auxiliary objectives, reducing FID by 35.2% and 33.3%, respectively, and producing substantially stronger motion-text alignment, as reflected by consistent gains in R-Precision, CLIP-score, and Matching score (Tables 1&2). These results demonstrate that the task meaningfully enhances the model’s ability to understand future-aware behavior semantics, which directly benefits the fidelity and coherence of the generated motion.
> >
> > Moreover, text is the modality through which intention is externally expressed. Human intentions are not directly observable. They are communicated via language. Since our model outputs linguistic expressions of intended future actions, evaluating these outputs using textual metrics ensures the predicted intention is semantically meaningful and coherent.
> >
> > ***Noting***: To avoid any potential conceptual ambiguity, we adopt reviewer 7Xxa’s suggestion and replace the term Intention Prediction with Future-Aware Behavior Understanding, which more precisely reflects the nature of our method.

---

> > > ### Author Response · Authors · 2025-11-25
> > > **Response to Reviewer 1tfi (3/3)**
> > >
> > > > **W4: Train/eval split for Mo440H and evaluator reproducibility.**
> > >
> > > Thank you for pointing out this missing clarification. We provide the full details below to ensure transparency and reproducibility.
> > >
> > > **(a) Mo440H data split**
> > >
> > > We adopt a fixed and deterministic split for all experiments:
> > >
> > > - 80% training
> > > - 5% validation
> > > - 15% test
> > >
> > > We will release the exact split indices in the camera-ready code repository so that all evaluators and models can be reproduced without ambiguity.
> > >
> > > **(b) Evaluator training**
> > >
> > > For HumanML3D dataset, we use the evaluator provided by MARDM. For Mo440H dataset, to prevent any data leakage:
> > >
> > > - The evaluator is trained strictly on the training split only.
> > > - The validation split is used solely for checkpoint selection.
> > > - The evaluator never accesses validation or test motions during training.
> > >
> > > Thus, the evaluation metrics are computed on motions that the evaluator has never seen, eliminating leakage concerns.
> > >
> > > **(c) Consistency with prior protocols**
> > >
> > > Our evaluator follows the exact MARDM evaluation protocol, including:
> > >
> > > - the same model architecture
> > > - identical optimization hyperparameters
> > > - the same feature extraction and metric computation pipeline
> > >
> > > This ensures direct comparability with prior T2M works.
> > >
> > > **(d) HumanML3D evaluation protocol**
> > >
> > > To avoid dataset contamination across benchmarks, we strictly isolate training sources:
> > >
> > > - For Table 1 (HumanML3D T2M evaluation), the model is trained only on HumanML3D.
> > > - For Mo440H experiments, we train a separate model exclusively on Mo440H.
> > >
> > > Thus, HumanML3D results do not benefit from Mo440H data, and vice versa.

---

> > > > ### Comment · Reviewer_1tfi · 2025-11-26
> > > >
> > > > I thank the authors for the detailed rebuttal and for clarifying several implementation details. The additional explanations improved my understanding of some parts of the pipeline, and I appreciate the effort that went into the response. That said, my overall assessment remains largely unchanged, and I still lean toward rejection, primarily because the core positioning and contribution of the work remain underspecified.
> > > >
> > > > Even after the rebuttal, I find it difficult to understand whether MoGIC is meant to be primarily:
> > > >
> > > > a dataset / benchmark for multi-modal motion generation,
> > > >
> > > > a method that leverages “intention understanding and visual context”,
> > > >
> > > > or both at once.
> > > >
> > > > Currently, it tries to do many things (L2M, V2M, multi-modal combinations, intention modeling, etc.) without a clearly stated, central research question. The different tasks (L2M, V2M, and their combinations) feel loosely connected, and there is no coherent protocol for evaluating how the different modalities jointly contribute to the same generation goal.
> > > >
> > > > On the dataset side, several points remain unclear or under-motivated:
> > > >
> > > > The purpose of the V2M task is not clearly articulated beyond “using visual context.” It is not obvious how V2M differs fundamentally from standard MoCap-based supervision, or how it complements L2M in a principled way. I still do not have a clear picture of what problem V2M uniquely enables us to study.
> > > >
> > > > It is still not fully clear from the rebuttal what exact dataset(s) the different components (L2M, V2M, IP head, etc.) are trained on, and how these splits align with the tables in the main paper. For instance, the L2M results reported in the rebuttal differ from the previous table, and the reason is not sufficiently explained in a way that reassures me about the overall experimental consistency.
> > > >
> > > > Downsampled video and FPS choice: The rebuttal emphasizes that the video is downsampled to 1 fps and “used as reference,” but this design choice is never systematically examined. If the video is meant to carry meaningful motion or visual context, why 1 fps instead of 0.5, 2, 3, or the full frame rate? How sensitive is the system to this? Without an analysis of different FPS settings, it is difficult to understand the actual contribution and necessity of the visual modality.
> > > >
> > > > There is no unified evaluation protocol that explicitly measures how text, video, and their combination trade off or complement each other for the same underlying motion target. As a result, it is hard to tell whether the multi-modal design really offers a robust advantage, or whether it just aggregates several loosely related settings.
> > > >
> > > > Overall, the dataset feels ambitious but somewhat “spread too thin”: it tries to cover many directions, yet the paper does not clearly explain what specific scientific question this dataset is best suited to answer.
> > > >
> > > > ---
> > > >
> > > > On the method side, the “intention” concept remains the central novelty claim, but it is still not precisely defined or justified:
> > > >
> > > > My original question about where the “intention” supervision comes from remains unresolved. The rebuttal does not clearly describe whether intention is explicitly annotated, derived from text, or inferred from motion in a principled way. Without a clear source of supervision, it is hard to consider “intention” as a well-defined, reproducible signal.
> > > >
> > > > Joint training with L2M: Since the IP head is jointly trained during L2M, the learned representation may primarily reflect the motion distribution, the text input, or some entangled mixture of both. As currently described, the “intention” vector appears very close to just being another encoding of the text input, which raises the question of what is actually gained beyond a standard text-based conditioning. From the pipeline figure, at least as I see it, the intention representation is derived from the same text that is already used as input. I understand that the pipeline is just for illustration, but identical text input and predicted intention is misleading. This makes the conceptual novelty of the intention module unclear: if intention ≈ text embedding, then the contribution is less about a new kind of supervision and more about an architectural rearrangement that the paper does not fully justify.
> > > >
> > > > The choice of “50% of the sequence” as a boundary for intention extraction seems arbitrary. There is no empirical or theoretical justification for why half of the motion is sufficient (or optimal) for capturing intention, especially given the diversity of human motions (where intention may be evident very early, midway, or only near the end). Without sensitivity analysis or a principled rationale, this design choice feels ad-hoc.
> > > >
> > > > Collectively, these issues make it difficult to accept the claim that the paper provides a clear and meaningful notion of “intention” that can be reused or built upon by the community. I therefore maintain my original score.

---

> > > > > ### Author Response · Authors · 2025-12-03
> > > > > **Response to Reviewer 1tfi (1/N)**
> > > > >
> > > > > **On the central coontribution of MoGIC**
> > > > >
> > > > > We thank the reviewer for raising this important point. We would like to clarify that MoGIC is not a collection of loosely connected tasks, but a unified multimodal motion generation framework designed around a single central research question:
> > > > >
> > > > > *How can a motion generator jointly model (i) future-aware behavioral structure and (ii) multimodal conditioning (language, vision, partial motion) so that the generated motion is coherent, and responsive to different input modalities?*
> > > > >
> > > > > The multiple tasks (L2M, V2M, V+L2M, future-aware behavior understanding, and motion captioning) are not independent objectives; rather, they are different manifestations of the same generative capability under different observed modalities. Importantly, these tasks are functionally coupled inside the architecture and benefit each other. For example:
> > > > >
> > > > > - Visual modality improves text-to-motion in two ways.
> > > > >
> > > > >     - Pretraining effect: In Stage-1, learning visual priors helps the backbone acquire stronger spatial reasoning and human-object interaction, which directly improve Stage-2 text-to-motion finetuning.
> > > > >     - Conditional effect: When used at inference time, the visual stream resolves semantic ambiguity in text descriptions (e.g., object location, moving distance), producing more precise motion generation.
> > > > >
> > > > > - The cross-modal backbone also enables strong performance on motion captioning. Because MoGIC learns a robust, modality-aligned representation in Stage-1, especially through the CMT module, the model can generate competitive textual descriptions after finetuning using only a 3-layer Transformer decoder, surpassing LLM-based baselines. This demonstrates that different tasks share and reinforce a common latent structure rather than being introduced independently.
> > > > >
> > > > > - Moreover, because we incorporat future-aware behavior understanding task in the first stage, MoGIC can output a complete behavior description based on partial action inputs without requiring fine-tuning.
> > > > >
> > > > > In addition, we would like to clarify that Mo440H was not created as an independent contribution, but as a necessary foundation for training MoGIC as a unified multimodal conditioned motion generator. Existing high-quality text-to-motion datasets (e.g., HumanML3D, KIT-ML) suffer from two fundamental limitations:
> > > > >
> > > > > - Lack of visual modality, which makes it impossible to train or evaluate models that require aligned video-motion-text inputs.
> > > > > - Insufficient data scale and diversity to support learning robust tri-modal representations.
> > > > >
> > > > > To enable MoGIC’s joint modeling of vision, language, and motion, we therefore integrate 21 publicly available datasets into Mo440H, perform unified preprocessing, and automatically annotate missing textual labels.

---

> > > > > > ### Author Response · Authors · 2025-12-03
> > > > > > **Response to Reviewer 1tfi (2/N)**
> > > > > >
> > > > > > **On the role and motivation of the V2M task**
> > > > > >
> > > > > > V2M is not positioned as a primary task in our framework. It serves as an exploratory extension that enables a class of applications difficult to support with language-only conditioning. For example, editing motions directly from video inputs generating visually grounded in-between motions (as demonstrated in the Figure 3). The motivation for introducing the visual modality is to compensate for the inherent limitations of language descriptions: text cannot reliably specify the precise location of interacted objects, the spatial layout of the scene, or the coarse trajectory of human joints. These forms of spatial and interaction knowledge are naturally captured by the visual stream, and their benefits are clearly reflected in our **LV2M** (no V2M) results.
> > > > > >
> > > > > > To further clarify this point, we have added more LV2M visualizations in Section A.3 of the supplementary material, showing how visual context complements language cues and leads to more consistent motion synthesis. Our goal is to allow users to leverage an easily accessible modality (visual input) to provide conditional information that is difficult or impossible to articulate through text alone.
> > > > > >
> > > > > > Within Mo440H, we partition the dataset into three subsets according to the available modalities:
> > > > > > - Mo440H-All: all sequences that contain motion data (this is equivalent to the full Mo440H dataset).
> > > > > > - Mo440H-ML: all motion-language paired samples.
> > > > > > - Mo440H-MLV: all samples containing motion, language, and video.
> > > > > >
> > > > > > During Stage-1 pretraining, we use Mo440H-All. For any missing modality, we simply mask out the corresponding tokens, ensuring that the model learns a unified representation across heterogeneous samples.
> > > > > >
> > > > > > - For fine-tuning and evaluation, we select the subset that matches the modalities required by each task:
> > > > > > - Tasks involving only motion and language, such as motion captioning, motion in-between, and language-to-motion generation, are trained and evaluated on Mo440H-ML.
> > > > > > - Tasks that require visual inputs, such as vision+language-to-motion generation, are trained and evaluated on Mo440H-MLV.
> > > > > >
> > > > > > The difference in the L2M results reported in the rebuttal arises because, following the reviewer’s request, we re-ran the experiments on the subsets which contain RGB videos (without the rendered mesh videos). This setting is not equivalent to the one used in the main paper, and therefore the numbers are not directly comparable.

---

> > > > > > > ### Author Response · Authors · 2025-12-03
> > > > > > > **Response to Reviewer 1tfi (3/N)**
> > > > > > >
> > > > > > > **On downsampled video and FPS choice**
> > > > > > >
> > > > > > > We thank the reviewer for raising this important question regarding the choice of 1 FPS and the sensitivity of MoGIC to different visual sampling rates. Below, we provide a detailed analysis and experimental evidence.
> > > > > > >
> > > > > > > **(1) Why 1 FPS?**
> > > > > > >
> > > > > > > The visual stream in MoGIC is not used as dense motion supervision; instead, it provides high-level semantic cues such as human-object spatial relations and coarse movement tendencies. Using dense visual frames causes the model to anchor on specific visual poses and behave more like deterministic keyframe interpolation, which reduces generative flexibility. Sparse visual cues (1 FPS) preserve semantic conditioning without suppressing motion diversity. We conducted an ablation comparing 1 FPS and 2 FPS inputs on the vision+language-to-motion benchmark (IDEA400). The results show that while 2 FPS slightly improves FID, it reduces diversity, indicating stronger visual over-conditioning.
> > > > > > >
> > > > > > > | FPS |   FID↓  | Top1↑ | Top2↑ | Top3↑ | Diversity↑ | Matching↓ |
> > > > > > > |-----|---------|--------|--------|--------|-------------|------------|
> > > > > > > | w/o vision | 0.646 | 0.280 | 0.459 | 0.582 | 9.111 | 3.428 |
> > > > > > > | 1 FPS | 0.443 | 0.285 | 0.466 | 0.588 | 9.106 | 3.321 |
> > > > > > > | 2 FPS | 0.423 | 0.284 | 0.469 | 0.597 | 8.903 | 3.226 |
> > > > > > >
> > > > > > > **(2) Dynamic-Aware Keyframe Experiments**
> > > > > > >
> > > > > > > To directly address the concern that 1 FPS might miss fast motion dynamics, we additionally evaluate motion-salient keyframes:
> > > > > > >
> > > > > > > - Optical flow peaks: top 10 frames with highest pixel-level motion
> > > > > > > - Joint acceleration peaks: top 10 frames with highest joint-level acceleration
> > > > > > >
> > > > > > > | Sampling Strategy |   FID↓  | Top1↑ | Top2↑ | Top3↑ | Diversity↑ | Matching↓ |
> > > > > > > |-------------------|---------|--------|--------|--------|-------------|------------|
> > > > > > > | w/o vision        | 0.646 | 0.280 | 0.459 | 0.582 | 9.111 | 3.428 |
> > > > > > > | 1 FPS             | 0.443 | 0.285 | 0.466 | 0.588 | 9.106 | 3.321 |
> > > > > > > | Optical-flow top-10 | 0.421 | 0.286 | 0.468 | 0.591 | 8.797 | 3.304 |
> > > > > > > | Acceleration top-10 | 0.418 | 0.285 | 0.468 | 0.593 | 8.817 | 3.302 |
> > > > > > >
> > > > > > > Both methods emphasize dynamic frames, but they lead to similar trade-offs: slightly better FID but reduced diversity.

---

> > > > > > > > ### Author Response · Authors · 2025-12-03
> > > > > > > > **Response to Reviewer 1tfi (4/N)**
> > > > > > > >
> > > > > > > > **On the choice of 50% of the sequence**
> > > > > > > >
> > > > > > > > We choose a 50% prefix because it offers strong supervision for future-aware behavior understanding. In the revised manuscript, we include an ablation comparing prefix ratios of 25%, 50%, 75%, and 100%. The results (Table 7, page 10) show that 50% yields the best downstream language-to-motion performance, achieving the lowest FID (0.134) and highest R-Precision Top-3 (0.926).
> > > > > > > >
> > > > > > > > | Visible Rate | Top1 ↑ | Top2 ↑ | Top3 ↑ | FID ↓  | Match ↓ | Div. ↑   |
> > > > > > > > |--------------|--------|--------|--------|--------|----------|-----------|
> > > > > > > > | 100%         | 0.646  | 0.845  | 0.919  | 0.198  | 1.910    | **12.623**   |
> > > > > > > > | 75%          | 0.648  | 0.847  | 0.922  | 0.171  | 1.905    | 12.599    |
> > > > > > > > | 50%          | **0.652**  | **0.851**  | **0.926**  | **0.134**  | **1.889**    | 12.434    |
> > > > > > > > | 25%          | 0.650  | 0.848  | 0.923  | 0.156  | 1.905    | 12.417    |
> > > > > > > >
> > > > > > > > This reflects a key trade-off intrinsic to future-aware modeling: a smaller prefix such as 25% provides insufficient contextual conditions to infer how the motion is likely to unfold, while a larger prefix (75-100%) leaves too little “future” to reason about, weakening the model’s ability to learn meaningful future-aware behavior patterns. Consequently, the 50% prefix is not an arbitrary design choice, it is the optimal balance between sufficient observed context and sufficient unobserved future for effective future-aware behavior understanding. This explanation and the new ablation results have been incorporated into the revised manuscript (Section 5.4 The proportion of visible motion prefixes in Future-Aware Behavior Understanding).

---

> ### Author Response · Authors · 2025-12-03
> **Response to Reviewer 1tfi (5/N)**
>
> **Concern on “Intention” (now Future-Aware Behavior Understanding)**
>
> We sincerely thank the reviewer for raising these important points. Following Reviewer 7Xxa’s suggestion, we have completely removed the previous “intention prediction” phrasing and replaced it throughout the manuscript with the more accurate term future-aware behavior understanding, accompanied by a full realignment of the conceptual narrative and methodological description.
>
> We clarify that future-aware behavior understanding is supervised by exactly the same motion-description annotations used in L2M. During the future-aware behavior understanding task, the model does not receive the full motion. It receives only a partial prefix of the trajectory (e.g., the first 50%), and must infer the high-level future behavior from incomplete observations. This supervisory setup is directly analogous to how future behavior must be inferred in real human motion understanding.
>
> A central misunderstanding seems to come from the pipeline figure. To avoid confusion, we explicitly note the following:
>
> - L2M and future-aware behavior understanding are trained with two separate forward passes and two separate losses.
> - The future-aware behavior understanding pass does not receive text as input, only the motion prefix.
> - Therefore, the learned representation cannot degenerate into a text-embedding replica, because the supervisory target is text describing future behaviors, while the input contains no textual information and no future motion.
>
> Conceptually, this task requires the model to learn structured reasoning over the motion prefix, not the text-based conditioning generation.
>
> **Why this is more than architectural rearrangement.**
>
> The value of the behavior-understanding module is not in adding another text projection, but in introducing a new supervision regime:
>
> - L2M learns mapping from language to full motion
> - Future-aware behavior understanding learns mapping from early motion to high-level future semantics
>
> Together, the two tasks enable the unified model to acquire behavior-level reasoning that improves generation quality.

---

### Official Review · Reviewer_LpQA · 2025-10-25

**Soundness:** 3
**Presentation:** 3
**Contribution:** 3
**Rating:** 6
**Confidence:** 3

**Summary:**

This paper addresses key limitations in text-conditioned human motion generation: the lack of explicit intention modeling and the inherent ambiguity of text-only descriptions. Its primary contributions are: **1) Explicit Intention Modeling**: It introduces a novel framework with disentangled generation heads: an Intention Prediction Head (IPH) that generates goal-oriented text, and a Motion Generation Head (MGH) that produces continuous motion trajectories. This allows the model to capture the underlying causal logic of actions. **2) Multi-Modal Conditioning**: It effectively incorporates the visual modality (low-frame-rate images) as a weak prior to resolve textual ambiguity and provide spatial context, enabling more precise and controllable motion generation. **3) Novel Architecture**: It proposes a Conditional Masked Transformer with a mixture-of-attention mechanism that dynamically aligns motion tokens with the most relevant text or visual tokens, handling temporal mismatches between modalities. **4) Large-Scale Benchmark**: It curates Mo440H, a large-scale dataset of 440 hours of motion data with text and visual annotations, facilitating future research.

Extensive experiments show that MoGIC achieves new state-of-the-art performance, significantly outperforming existing methods on metrics like FID and R-Precision.

**Strengths:**

The main strengths of this paper are:

**Novel Idea**: It introduces a smart new concept instead of just making motions from text. This helps the AI understand why a motion happens, not just how it looks.

**Multi-modal Power**: It successfully combines text with visual information (images). This solves the problem of text being vague and makes the generated motions much more precise and realistic.

**Strong Results**: The method clearly outperforms all current state-of-the-art models on standard tests. The improvements in metrics like FID are significant.

**Great Flexibility**: The same model can handle many different tasks, like generating motion from text, from images, completing missing motion, and even explaining the intention behind a motion.

**Valuable Resource**: It creates a large, new dataset (Mo440H), which will be very useful for other researchers in the field.

**Weaknesses:**

***The incorporation of the visual modality is a commendable aspect of this work***. However, the proposed strategy of using a fixed, low-frame-rate (1 fps) sampling as the sole visual conditioning signal is a significant limitation that is not adequately addressed. This choice undermines the potential of visual control in several critical ways:

1).**Lack of Adaptability to Motion Dynamics**: A fixed 1 fps sampling rate is inherently agnostic to the inherent speed and dynamics of the target motion. This is a critical flaw.

For slow, sustained actions (e.g., "meditating"), 1 fps may be sufficient, even over-sampled.

For fast, transient actions (e.g., "a quick jab," "a golf swing," "clapping hands"), a 1 fps sampling is highly likely to miss the crucial kinematic phase of the action (e.g., the moment of impact). As the authors rightly noted, language cannot specify fine-grained spatiotemporal details; unfortunately, a rigid 1 fps visual stream often fails to capture them as well. This can lead to generated motions that are plausible in isolation but temporally misaligned or physically implausible when precise timing is required.

2). **Ignition of the "Keyframe" Paradigm for Controllability**: The current method treats visual inputs as a passive, weak prior. For true controllability, especially in downstream applications like game development or animation, artists and developers require an active, intentional control mechanism. The ability to provide sparse, user-defined keyframes is the industry standard for a reason: it allows for precise control over timing and pose.

The authors' model architecture, with its mixture-of-attention mechanism, seems capable of handling such sparse keyframes, but the training paradigm does not explore this. The work would be significantly strengthened by investigating visual conditioning on semantically meaningful keyframes (e.g., start, end, and apex of a jump) rather than arbitrarily sampled low-frame-rate images.

***Furthermore, the visual evidence supporting the core claim is severely lacking***. The solitary qualitative example in Figure 3 is insufficient. For instance, in a rapid "weightlifting" motion, if the sampled frames coincidentally miss the key "lifting" phase and only show the barbell on the ground, the visual condition would actively contradict the text description. It remains unproven whether the model can robustly override such deceptive visual cues and rely on the stronger textual prior or learned motion, or if it would generate an incorrect sequence. Furthermore, the qualitative analysis supporting the core claim is severely lacking. In the supplementary material, only provide many visualization results of language-to-motion generation, lacking VL2M results.

**Questions:**

1. The paper demonstrates strong performance on in-distribution datasets. However, how does the model generalize to out-of-distribution (OOD) descriptions or intentions? For example, if the text describes a novel combination of actions ("do a handstand while hopping on one foot") that is unlikely to be in the training data, can the model still generate a plausible motion? An analysis of the model's performance on OOD or compositional tasks would greatly strengthen the claims about its generalizability.

2. The construction of Mo440H is a contribution, but it raises concerns about quality and bias. A significant portion of the textual descriptions is generated automatically by Qwen2.5-VL. What measures were taken to control for the inherent biases and potential errors of this large vision-language model? Furthermore, have you analyzed the distribution of action types within Mo440H? Is there a risk that the model's performance is biased towards the most frequent action categories in this aggregated dataset?

---

> ### Author Response · Authors · 2025-11-26
> **Response to Reviewer LpQA (1/5)**
>
> We sincerely thank the reviewers for their constructive feedback. We appreciate the time and effort invested in reading our manuscript and providing valuable suggestions. We have carefully addressed comments below.
>
> > **W1: Lack of Adaptability to Motion Dynamics**
>
> We sincerely thank the reviewer for raising this important point. The concern about whether sparsely sampled visual frames can capture motion dynamics is highly valid, and we address it comprehensively below.
>
> **(1) Why we choose 1 FPS videos as vision inputs**
>
> Our use of 1 FPS visual inputs is based on the role that the visual modality plays in our formulation. The goal of the visual stream is not to provide dense kinematic supervision or reconstruct fast transient motion phases. Instead, it serves as a sparse, high-level semantic cue, offering information such as environment context, human–object interaction type, and movement tendencies. These signals are integrated through our adaptive adaptive top-k cross-attention, which is intentionally designed to avoid over-conditioning on per-frame timing. If the visual modality is made dense or overly motion-centric, such as by sampling all high-dynamics frames, the model becomes anchored to specific visual poses and tends to behave like keyframe interpolation rather than semantic motion generation. This undermines generative flexibility, which is a central goal of our work.
>
> **(2) Experiments on higher FPS settings**
>
> To verify whether denser visual sampling affects performance, we conducted an ablation with 2 FPS inputs on IDEA400 (vision+language-to-motion). The results are shown below:
>
> | FPS            | FID↓             | Top1↑            | Top2↑            | Top3↑            |   Diversity↑    |  Matching↓      |
> |-----------------------------------|-----------------|-----------------|-----------------|-----------------|-----------------|-----------------|
> | 1 | 0.443 | **0.285** |0.466 | 0.588 | **9.106** | 3.321 |
> | 2 | **0.423** | 0.284 | **0.469** | **0.597** | 8.903 | **3.226** |
>
> Although the 2 FPS condition mildly improves FID, it also leads to a noticeable drop in motion diversity. This supports our design principle that sparse visual priors preserve generative flexibility. Moreover, 1 fps videos already capture the major trends of joint-level dynamics (e.g., the direction and relative magnitude of limb movement) in most cases, which are sufficient for high-level conditioning even if they do not encode exact, frame-accurate velocities. For a model whose goal is motion generation rather than reconstruction, these trends are the primary useful signals.
>
> **(3) Dynamic -Aware Keyframe Sampling**
>
> To directly address the reviewer’s concern regarding motion dynamics, we performed additional experiments using motion-aware keyframe selection. We extracted visually dynamic frames using
>
> - Optical flow magnitude, selecting the 10 frames with the highest pixel-level motion.
> - Joint-acceleration peaks, selecting the 10 frames with the largest joint acceleration.
>
> Both methods successfully select frames that emphasize fast or meaningful motion phases.
>
> | Sampling            | FID↓             | Top1↑            | Top2↑            | Top3↑            |   Diversity↑    |  Matching↓      |
> |-----------------------------------|-----------------|-----------------|-----------------|-----------------|-----------------|-----------------|
> | w/o vision | 0.646 | 0.280 | 0.459 | 0.582 | **9.111** | 3.428 |
> | 1 FPS | 0.443 | **0.285** | 0.466 | 0.588 | 9.106 | 3.321 |
> | Optical flow-based  | 0.421 | **0.286** | **0.468** | 0.591 | 8.797 | 3.304 |
> | Acc.-based | **0.418** | 0.285 | **0.468** | **0.593** | 8.817 | **3.302** |
>
>
> The results consistently revealed the same trade-off as the FPS study: while these motion-aware sampling strategies slightly reduce FID, they also suppress motion diversity because the model becomes more tightly constrained by deterministic visual poses. Importantly, 1 fps strategy already offers a substantial improvement over the “w/o vision” baseline, demonstrating that sparse visual inputs are sufficient to capture the high-level vision semantic that our model requires, even without explicitly aligning to fine-grained dynamics.
>
> Although experimentally and motivationally we believe that 1 FPS meets our needs, we fully agree with the reviewer that attending to motion-dynamic frames is highly valuable, especially for vision-based motion editing, generation, reconstruction and control tasks where capturing fine-grained temporal transitions is crucial. This insight highlights an important direction for our future work.

---

> ### Author Response · Authors · 2025-11-26
> **Response to Reviewer LpQA (2/5)**
>
> > **W2: On the use of sparse visual frames vs. keyframe-based controllability**
>
> We thank the reviewer for this insightful suggestion. We fully agree that user-defined, semantically meaningful keyframes represent an ideal form of controllable motion edit and generation. However, our work targets a different goal and problem formulation. The vision modality aims at providing weak vision priors/conditions, e.g. approximate location of objects and coarse joint moving trend, rather than carefully curated keyframes. Moreover, the Mixture-of-Attention with Adaptive Scope mechanism also enables the model to capture high-level visual semantics without requiring users or annotators to specify aligned key poses.
>
> Adopting sparse, semantically curated keyframes would fundamentally shift the task toward controllable motion generation, where the visual input explicitly encodes target poses and timing. In contrast, our work focuses on reducing multimodal ambiguity and injecting high-level visual priors, rather than enforcing pose-level constraints. We fully agree that meaningful keyframe conditioning is a valuable direction, and our mixture-of-attention architecture has potential to support it. This naturally connects to our planned future work, where we intend to explore spatial constraints, automatic keyframe discovery, and stronger visual-intent control for animation and editing applications.
>
> > **W3: Visual signals may be misleading & Visual evidence is insufficient**
>
> Thank you for highlighting this concern. We will explain this situation below:
>
> **(1) Text dominates vision**
> Our results show that language provides a substantially stronger conditioning signal than vision: L2M achieves FID 0.330 vs. 0.634 for V2M (Table 5). This significant gap indicates that when cross-modal information conflicts, the model naturally relies more heavily on the textual prior.
>
> **(2) Robustness to Conflicting Visual Cues**
> Importantly, the model is structurally equipped to handle potentially misleading visual frames. The adaptive Top-K attention mechanism selectively focuses on informative tokens and consistently suppresses unreliable tokens during training, preventing cases where sparse or missing key motion phases could distort generation.
>
> **(3)	Qualitative Results**
> Thank you for pointing out the lack of the qualitative results. In the updated PDF, we will include more VL2M visualizations together with attention-map examples to support our claim.

---

> ### Author Response · Authors · 2025-11-26
> **Response to Reviewer LpQA (3/5)**
>
> > **Q1: Generalization to OOD Descriptions**
>
> We thank the reviewer for raising this important question regarding OOD and compositional generalization. We conduct two dedicated zero-shot experiments to directly evaluate MoGIC's performance under out-of-distribution data.
>
> **(1) Zero-shot generalization on an unseen dataset (IDEA400)**
>
> We trained MoGIC and MARDM (for comparisons) under two settings:
>
> - HumanML3D only
> - Mo440H (except for IDEA400 subset)
>
> Then, both models were evaluated zero-shot on the unseen IDEA400 dataset. The comparison results as shown as below:
>
> | Method    | Train data        | FID↓             | Top1↑            | Top2↑            | Top3↑            |   Diversity↑    |  Matching↓      |
> |------------------|-----------------|-----------------|-----------------|-----------------|-----------------|-----------------|-----------------|
> MARDM | HumanML3D | 13.968 | 0.134 | 0.215 | 0.328 | 7.654 | 5.669 |
> MoGIC | HumanML3D | **9.912** | **0.156** | **0.248** | **0.360** | **7.885** | **5.369** |
> MARDM | Mo440H-ML w/o IDEA400 | 6.344 | 0.228 | 0.374 | 0.502 | 10.221 | 3.467 |
> MoGIC | Mo440H-ML w/o IDEA400 | **4.269** | **0.242** | **0.403** | **0.541** | **10.747** | **3.349** |
>
>
> MoGIC consistently outperforms MARDM across all zero-shot settings, reducing FID by an average of 30.85%. This indicates that our method learns motion semantics that transfer beyond the training distribution.
>
> **(2)	Zero-shot compositionality under unseen, fine-grained descriptions**
>
> To further test OOD textual generalization, we use FineMotion’s re-annotations of HumanML3D:
>
> - The motions are the same as HumanML3D,
> - But the text descriptions are much more fine-grained and stylistically very different (joint-level descriptions instead of high-level summaries).
>
> We train our method and MARDM on HumanML3D and evaluate them using text descriptions provided by FineMotions:
>
> | Method    | FID↓             | Top1↑            | Top2↑            | Top3↑            |   Diversity↑    |  Matching↓      |    CLIP score↑ |
> |------------------|-----------------|-----------------|-----------------|-----------------|-----------------|-----------------|----------------|
> MARDM | 35.688 | 0.075 | 0.141 | 0.192 | **7.001** | 6.240 | 0.402 | 0.402 |
> MoGIC | **30.74** | **0.087** | **0.157** | **0.220** | 6.985 | **5.170** | **0.497** |
>
> MoGIC shows better FID and alignment with fine-grained descriptions (higher CLIP, R-precision and lower Matching Score), despite never having seen this annotation style during training. However, the FID results and alignment metrics were not ideal (one reason is that the evaluator had not seen these texts during training.), indicating that although our method outperforms other SOTA methods (MARDM), achieving robust results on unseen style descriptions remains challenging.

---

> > ### Author Response · Authors · 2025-11-26
> > **Response to Reviewer LpQA (4/5)**
> >
> > > **Q2: Bias control and reliability of Qwen2.5-VL–generated captions & Distribution of action types**
> >
> > We appreciate the reviewer’s concern. Below we clarify the design, validation, and bias control of Mo440H’s auto-generated captions.
> >
> > **(1) Bias control and reliability of Qwen2.5-VL–generated captions**
> >
> > - **All baselines are trained on exactly the same labels.**
> >
> >     No model has privileged access to better text. Thus, the performance gap reflects architectural differences rather than label source bias.
> >
> > - **Our improvements hold on human-labeled benchmarks as well.**
> >
> >     On the human-annotated dataset HumanML3D (Table 1), MoGIC still achieves substantial gains (FID −36.8%, R-Precision Top3 +4%, Matching −8.3%), confirming that the gains are not tied to Qwen-style text.
> >
> > - **Filtering pipeline**
> >
> >     Our filtering pipeline applied three safeguards to avoid VLM-induced errors.
> >     - **Dataset-level rejection**: We removed datasets whose visual signals are too sparse or discontinuous to support reliable captioning (e.g., Chairs).
> >     - **Model-based rejection**: An early generator trained on auto-captions was used to detect data sources that degrade motion quality; PROX was excluded due to severe occlusions despite seemingly correct captions.
> >     - **Manual verification**: For the remaining mocap-style datasets, we manually inspected random samples and observed a low error rate, confirming their suitability.
> >
> > - **Human evaluation confirms Qwen-generated labels are reliable.**
> >
> >     We conducted a controlled human study to assess the quality of Qwen-generated captions. We randomly sampled 200 Mo440H sequences, manually re-annotated them, and asked 10 independent evaluators to perform blind A/B comparisons. Each rater scored the two captions on:
> >     - Description accuracy (0–5): whether the caption correctly describes the motion sequence
> >     - Linguistic coherence & fluency (0–5)
> >     - Detail richness (0–5): whether the description is detailed and small but relevant movement details are captured.
> >
> >     Raters were not told which caption came from humans or from Qwen. The results are shown in this table.
> >
> >     | Source | Description accuracy↑ | Linguistic coherence↑ | Detail richness↑ |
> >     |-----------|------------------------------|-----------------------------|---------------------|
> >     | Human-annotated | **4.05** | **4.10** | 3.58 |
> >     | Qwen-annotated | 3.84 | 4.06 | **4.18** |
> >
> >     Qwen-generated captions are at least comparable to human-written text in accuracy and fluency, and significantly more detailed, confirming their suitability as training supervision.
> >
> > - **Zero-shot validation on a purely human-labeled dataset (HuMMan).**
> >
> >     To test whether the bias introduced by the Qwen-annotated dataset would affect the generation capability of MoGIC, we train two separate versions:
> >
> >     - Model A: trained on all datasets except HuMMan (includes Qwen-generated + human-annotated labels).
> >     - Model B: trained only on all human-annotated datasets except HuMMan, including HumanML3D, HIMO, and InterHuman (no Qwen-generated labels involved).
> >
> >     Both models were evaluated zero-shot on HuMMan, which is entirely human-annotated and unseen during training.
> >
> >     | Train dataset | Test dataset | FID↓ | Top1↑ | Top2↑ | Top3↑ | Diversity↑ | Matching↓ |
> >     |-|-|-|-|-|-|-|-|
> >     | HumanML3D+HIMO+InterHuman | HuMMan | 29.912 | 0.046 | 0.090 | 0.122 | 3.868 | 9.958 |
> >     | All w/o HuMMan | HuMMan | **13.544** | **0.077** | **0.150** | **0.207** | **6.422** | **7.532** |
> >
> >     We observe that adding more Qwen2.5-VL-generated annotations significantly boosts zero-shot performance on the human-labeled HuMMan dataset (FID: 29.91 → 13.54). This shows that Qwen-generated labels serve as effective supervisory signals that improve true generalization.

---

> > > ### Author Response · Authors · 2025-11-26
> > > **Response to Reviewer LpQA (5/5)**
> > >
> > > **(2) Distribution of action types**
> > >
> > > Mo440H is an open-vocabulary motion dataset in which each sequence often contains multiple actions that may appear sequentially, concurrently (e.g., “walk while waving”), or with repeated sub-patterns. Because these actions can be combined or overlap in time, it is difficult to define a closed and mutually exclusive set of “action types’’ that meaningfully covers all cases. Moreover, different textual expressions may refer to the same underlying motion (e.g., “raise arms,” “lift both arms,” “extend elbows upward”), making discrete action counting less reliable. Therefore, action-type are not straightforward to compute in a principled way.
> > >
> > > Alternatively, we computed verb-frequency statistics across all Mo440H captions. Here shows the 10 most frequently occurring words:
> > >
> > > |         | walk | stand | move | extend | raise | bend | remain | lift | turn | lower |
> > > |----------------|------|-------|-------|--------|--------|-------|---------|--------|--------|--------|
> > > | Counter     | 65,945 | 64,238 | 41,958 | 41,384 | 39,398 | 35,612 | 30,696 | 23,891 | 21,518 | 18,926 |
> > > | Frequency | 5.84% | 5.69% | 3.72% | 3.67% | 3.49% | 3.16% | 2.72% | 2.12% | 1.91% | 1.68% |
> > >
> > > While common verbs appear more often, Mo440H contains substantial long-tail diversity actions (2000+ types) and each motion description typically contains multiple verbs, preventing the model from relying on high-frequency descriptions. Moreover, MoGIC achieves the best performance on the Mo440H and human-labeled HumanML3D dataset, and shows better generalization to IDEA400, FineMotion and HuMMan, indicating that it does not overfit frequent actions but instead learns robust motion semantics.

---

> > > > ### Comment · Reviewer_LpQA · 2025-11-27
> > > >
> > > > Thank you for the authors' efforts in responding to my comments. My major concerns have been adequately addressed, and I acknowledge the improvements made to the manuscript. While I maintain my view that the number of visualizations is a relative weakness, it is not significant enough to change my overall positive assessment of this work. Therefore, I will maintain my original score.

---

> > > > > ### Author Response · Authors · 2025-12-03
> > > > > **Second Round Response to Reviewer LpQA**
> > > > >
> > > > > We sincerely appreciate the reviewer’s positive assessment and thoughtful follow-up.
> > > > >
> > > > > Regarding the visualization issue, we have expanded the visualization section in the updated supplementary material. In particular, we **added a new set of results focusing on vision+language-to-motion and its direct comparison with language-only generation.** These examples are accompanied by detailed analysis to illustrate how visual context resolves textual ambiguity and improves motion plausibility.
> > > > >
> > > > > Please refer to Section A.3 (VISUALIZATION RESULTS) and Figure A6, where we present these additional qualitative comparisons.
> > > > >
> > > > > We are grateful for your constructive feedback, which has helped us significantly improve the clarity and completeness of the paper.

---

### Official Review · Reviewer_7Xxa · 2025-10-31

**Soundness:** 3
**Presentation:** 2
**Contribution:** 3
**Rating:** 4
**Confidence:** 5

**Summary:**

This paper proposes MoGIC, a unified framework for motion generation that integrates "intention understanding" and "visual context." The method models intention via an "Intention Prediction Head" (IPH) that predicts text from motion prefixes, and it uses sparse visual frames to disambiguate text. The method achieves a strong SOTA performance (FID 0.070) on the HumanML3D benchmark and contributes a new auto-annotated dataset, Mo440H.

**Strengths:**

- The paper's most significant contribution is its SOTA performance on HumanML3D. A huge FID reduction is a very impressive achievement and strongly demonstrates the method's effectiveness.

- The proposed "intention loss" (predicting full text from a motion prefix) is a novel and effective auxiliary training objective. The ablation study clearly shows this predictive task is a stronger supervisory signal than a standard M2T captioning loss.

- The use of sparse, weakly-aligned visual inputs to resolve the ambiguity of text-only descriptions (e.g., the weightlifting example in Fig 3) is a reasonable and practical contribution, confirmed by ablations in Table 5.

**Weaknesses:**

- The paper's central narrative is built on a foundation of **significant overclaiming**. The "intention understanding" and "causal logic" repeatedly mentioned are, in practice, a proxy task: "predicting a full text description from the first 50% of a motion sequence." Equating this M2T task with high-level cognitive intent or "internal causal structure" is a major conceptual leap without evidence. The attempt to differentiate this from standard M2T methods (like MotionGPT) by calling one "mapping" and this one "intention" is exaggerated and.

- The new Mo440H dataset's text labels are auto-generated by a very strong, contemporary VLM (Qwen2.5-VL-Max). Claiming SOTA performance (especially on captioning) on a dataset whose labels were synthesized by a SOTA model is problematic. The model's performance may be heavily reliant on these high-quality synthetic labels rather than the superiority of the model itself.

**Questions:**

- Given that the IPH is functionally a "prefix-to-text" M2T predictor, will the authors please substantially tone down the claims about "cognitive intention," "goals," and "causal logic" in the final version? The task's effectiveness as an auxiliary loss is clear (Strength 2), but the narrative is overblown.

- Can the authors discuss the potential impact of using a SOTA VLM to generate labels on the results (especially for captioning)? How can we be sure that the captioning SOTA on Mo440H isn't just a result of the model being good at "mimicking" the Qwen-VL's output style?

---

> ### Author Response · Authors · 2025-11-24
> **Response to Reviewer 7Xxa (1/2)**
>
> We sincerely thank the reviewers for their constructive feedback. We appreciate the time and effort invested in reading our manuscript and providing valuable suggestions. We have carefully addressed comments below.
>
> > W1: Overclaiming of the Intention Prediction & Q1: Toning down the claims
>
> We thank the reviewer for the thoughtful comments and for pointing out the potential risk of overinterpretation in our narrative. We understand that phrases such as “intention understanding” or “causal logic” may give an impression of stronger claims than what we aimed to express. However, the term "cognitive intention" did **not** appear in our article. Our intention was to describe the **future-aware behavior semantic** implied by the early part of the motion rather than psychological or cognitive constructs.
>
> Importantly, the reviewer’s interpretation that our Intention Prediction is “merely a prefix-to-text M2T predictor” does **not** fully reflect the nature of the task:
>
> - Existing M2T models (e.g., MotionGPT) reconstruct captions from fully-observed motion, where the mapping is primarily descriptive.
> - Our Intention Prediction task removes the entire second half of the motion sequence and requires the model to infer future-aware behavior semantics (in text format) that must be consistent with the unobserved future part.
> - This makes the task fundamentally different from caption reconstruction. **The model does require learning a structural dependency between early motion and logically consistent future-aware behavior semantics, which we believe is a meaningful form of causal reasoning within the motion domain.**
>
> This distinction is further validated by the ablation results in Table 2.
>
> | Methods                           | Top1            | Top2            | Top3            | FID↓            | Matching↓       | Diversity↑      |
> |-----------------------------------|-----------------|-----------------|-----------------|-----------------|-----------------|-----------------|
> | MoGIC *only L2M loss*      | 0.637   | 0.836   | 0.908   | 0.201   | 2.003   | 12.392  |
> | MoGIC *L2M + Caption loss* | 0.646   | 0.845   | 0.919   | 0.198   | 1.910   | **12.623**  |
> | MoGIC *L2M + Int. loss*  | **0.652** | **0.851** | **0.926** | **0.134** | **1.889** | 12.434  |
>
> When we introduce the Intention Prediction loss (MoGIC *L2M + Int. Loss*), **the FID drops by 33.3%** relative to the baseline (from 0.201 → 0.134). In contrast, adding a captioning loss (MoGIC *L2M + Caption. Loss*), which only reconstructs full-sequence text, yields a negligible 1.5% reduction (from 0.201 → 0.198).
>
> Such a large disparity in motion-quality improvement indicates that the Intention Prediction task is not only functioning as a caption-reconstruction loss. If both tasks were equivalent in nature, their impact on motion metrics would be comparable. Instead, Intention Prediction produces large FID improvement, demonstrating that its benefit arises from future-aware behavior understanding rather than text summarization.
>
> That said, we fully understand how the current wording (e.g., “intention understanding”) could be interpreted more strongly than we intended. We will revise the narrative to avoid ambiguity and to more accurately describe what our model does: **future-aware behavior understanding.**

---

> ### Author Response · Authors · 2025-11-24
> **Response to Reviewer 7Xxa (2/2)**
>
> > W2 & Q2: On using Qwen2.5-VL-Max to synthesize Mo440H captions
>
> We thank the reviewer for raising this important concern. We respectfully clarify that the concern does not affect the validity of our comparisons, for several reasons:
>
> - **All baselines are trained on exactly the same labels.**
>
>     No model has privileged access to better text. Thus, the performance gap reflects architectural differences rather than label source bias.
>
> - **Our improvements hold on human-labeled benchmarks as well.**
>
>     On the human-annotated dataset HumanML3D (Table 1), MoGIC still achieves substantial gains (FID −36.8%, R-Precision Top3 +4%, Matching −8.3%), confirming that the gains are not tied to Qwen-style text.
>
> - **Using a strong VLM for annotation is now standard practice in motion datasets.**
>
>     Manual labeling is costly and often lacks fine-grained detail. Many large-scale motion datasets rely partly or entirely on modern VLMs (e.g. MotionX, Go-to-Zero). Moreover, Mo440H also contains human-labeled subsets (HumanML3D, HIMO, HuMMan, InterHuman), ensuring diverse annotation styles.
>
> - **Human evaluation confirms Qwen-generated labels are reliable.**
>
>     We conducted a controlled human study to assess the reliability of Qwen-generated captions. Specifically, we randomly sampled 200 motion sequences from Mo440H and re-annotated each sequence manually. Then, we asked 10 independent human evaluators (not involved in our project) to perform blind A/B comparisons between the human-written captions and the Qwen-generated captions. Each rater scored the two captions:
>     - Description accuracy (0–5): whether the caption correctly describes the motion sequence
>     - Linguistic coherence & fluency (0–5)
>     - Detail richness (0–5): whether the description is detailed and small but relevant movement details are captured.
>
>     Raters were not told which caption came from humans or from Qwen. The results are shown in this table.
>
>     | Source | Description accuracy↑ | Linguistic coherence↑ | Detail richness↑ |
>     |-----------|------------------------------|-----------------------------|---------------------|
>     | Human-annotated | **4.05** | **4.10** | 3.58 |
>     | Qwen-annotated | 3.84 | 4.06 | **4.18** |
>
>     Qwen-generated captions are at least comparable to human-written text in accuracy and fluency, and significantly more detailed, confirming their suitability as training supervision.
>
> - **Zero-shot validation on a purely human-labeled dataset.**
>
>     To test whether MoGIC is simply learning to mimic Qwen’s writing style, we trained two separate versions:
>
>     - Model A: trained on all datasets except HuMMan (includes Qwen-generated + human-annotated labels).
>     - Model B: trained only on all human-annotated datasets except HuMMan, including HumanML3D, HIMO, and InterHuman (no Qwen-generated labels involved).
>
>     Both models were evaluated zero-shot on HuMMan, which is entirely human-annotated and unseen during training.
>
>     | Train dataset | Test dataset | FID↓ | Top1↑ | Top2↑ | Top3↑ | Diversity↑ | Matching↓ |
>     |-|-|-|-|-|-|-|-|
>     | HumanML3D+HIMO+InterHuman | HuMMan | 29.912 | 0.046 | 0.090 | 0.122 | 3.868 | 9.958 |
>     | All w/o HuMMan | HuMMan | **13.544** | **0.077** | **0.150** | **0.207** | **6.422** | **7.532** |
>
>     We observe that adding more Qwen2.5-VL–generated annotations significantly boosts zero-shot performance on the human-labeled HuMMan dataset (FID: 29.91 → 13.54). This shows that Qwen-generated labels serve as effective supervisory signals that improve true generalization, not as stylistic cues that the model overfits to.

---

> > ### Comment · Reviewer_7Xxa · 2025-11-25
> > **Unresolved concerns regarding the arbitrary definition of "Intention" and systemic overclaiming**
> >
> > I appreciate the authors' effort in providing additional experiments, particularly the zero-shot validation on HuMMan, which helps alleviate concerns regarding the reliance on synthetic labels. However, after carefully considering the rebuttal, I remain unconvinced regarding the paper's core conceptual framing. I cannot recommend acceptance in its current form for the following reasons:
> >
> > **The Arbitrariness of the "50%" Definition**.
> >
> > The authors argue that the "Intention Prediction" task is meaningful because it predicts future semantics from an incomplete sequence. However, the operational definition of "intention" relies heavily on an arbitrary threshold: truncating the latter 50% of the motion.
> >
> >
> > - Lack of Basis: There is no theoretical or empirical justification provided for why 50% represents "intention." Why not 20%, 30%, or the first 10 frames?
> >
> > - Action vs. Intention: In many motion sequences (e.g., "jumping," "kicking," or "throwing"), the action is physically explicit and fully recognizable by the 50% mark. At this stage, the model is performing "early action recognition" or "motion completion," not inferring a latent "cognitive intention" or "causal structure"  that precedes the action.
> >
> > The rebuttal does not address why this specific engineering choice (50% prefix) justifies the grand claim of understanding "human goals" or "causal logic".
> >
> >
> > **Systemic Overclaiming Cannot be Fixed by Minor Wording Changes**. The authors proposed to tone down terms like "intention understanding" to "future-aware behavior understanding." While this is a step in the right direction, the issue is systemic, not merely lexical.
> >
> > - The entire paper—from the title ("Intention Understanding"), the Abstract ("uncovers latent human goals" ), the Introduction ("internal causal structure" ), to the module names ("Intention Prediction Head")—is built around this narrative.
> >
> >
> > - The "intention" concept is not just a label; it is the central storytelling device used to differentiate this work from other M2T/prefix-completion methods.
> >
> > - Changing this narrative requires a fundamental restructuring of the paper, including the motivation, method description, and discussion, which goes beyond the scope of a revision during the rebuttal phase.
> >
> > While I acknowledge the method achieves good empirical results (SOTA FID), the scientific packaging of the method is misleading. Equating a "50% prefix-to-text" task with "understanding causal logic/intention" is a conceptual leap that is not rigorously supported.
> >
> > Therefore, I maintain my rating. I strongly suggest the authors fundamentally revise the manuscript to align the claims with the actual engineering contribution (i.e., a robust prefix-based auxiliary task for consistency) and resubmit to a future venue. The current narrative framing is too far removed from the technical reality.

---

> > > ### Author Response · Authors · 2025-12-03
> > > **Second Round Response to Reviewer 7Xxa (1/2)**
> > >
> > > We thank the reviewer again for the thoughtful follow-up comments. We address the raised concerns point-by-point and highlight the additional analyses and structural revisions already included in our updated manuscript.
> > >
> > > **1. On the “Arbitrariness” of the 50% Prefix Definition**
> > >
> > > We choose a 50% prefix because it offers strong supervision for future-aware behavior understanding. In the revised manuscript, we include an ablation comparing prefix ratios of 25%, 50%, 75%, and 100%. The results (Table 7, page 10) show that 50% yields the best downstream language-to-motion performance, achieving the lowest FID (0.134) and highest R-Precision Top-3 (0.926).
> > >
> > > | Visible Rate | Top1 ↑ | Top2 ↑ | Top3 ↑ | FID ↓  | Match ↓ | Div. ↑   |
> > > |--------------|--------|--------|--------|--------|----------|-----------|
> > > | 100%         | 0.646  | 0.845  | 0.919  | 0.198  | 1.910    | **12.623**   |
> > > | 75%          | 0.648  | 0.847  | 0.922  | 0.171  | 1.905    | 12.599    |
> > > | 50%          | **0.652**  | **0.851**  | **0.926**  | **0.134**  | **1.889**    | 12.434    |
> > > | 25%          | 0.650  | 0.848  | 0.923  | 0.156  | 1.905    | 12.417    |
> > >
> > > This reflects a key trade-off intrinsic to future-aware modeling: a smaller prefix such as 25% provides insufficient contextual conditions to infer how the motion is likely to unfold, while a larger prefix (75-100%) leaves too little “future” to reason about, weakening the model’s ability to learn meaningful future-aware behavior patterns. Consequently, the 50% prefix is not an arbitrary design choice, it is the optimal balance between sufficient observed context and sufficient unobserved future for effective future-aware behavior understanding. This explanation and the new ablation results have been incorporated into the revised manuscript (Section 5.4 The proportion of visible motion prefixes in Future-Aware Behavior Understanding).
> > >
> > >
> > > **2. On “Action Recognition” vs. “Future-Aware Behavior Understanding”**
> > >
> > > We appreciate the reviewer’s concern that some actions may already be observable by the halfway point. However, Mo440H contains a wide variety of multi-stage, interaction-rich, and causally structured behaviors, far beyond simple atomic motions such as jumping or kicking (in fact, such isolated atomic actions are rare in both Mo440H and HumanML3D).
> > > For example, Mo440H integrates multiple human-object interaction datasets (e.g., IMHD, HIMO), where a motion sequence typically consists of a chain of causally linked interactions. Several simple examples (can be found in our updated supplementary materials) include:
> > >
> > > - A person shifts a chair → they are highly likely to sit down afterward.
> > > - A person lifts and carries a box → they usually move it to another location and place it down.
> > >
> > > Moreover, numerous human-human interactions and multi-step personal motions exhibit equally rich future dependencies.
> > >
> > > In most sequences, the first 50% provides strong cues but does not reveal the full future, meaning the model must infer what plausible behavior will unfold next. Therefore, the proposed future-aware behavior reasoning task goes far beyond recognizing “what is happening now”; it explicitly requires predicting the most semantically coherent future behavior pattern.
> > >
> > > Our ablation study provides strong empirical evidence for this distinction (Table 2). Replacing the future-aware reasoning task with a caption reconstruction task significantly degrades performance, with FID worsening from 0.134 (MoGIC L2M + Und. loss) to 0.198 (MoGIC L2M + Caption loss) and clear declines across all text-motion alignment metrics. The magnitude of this performance gap indicates that the proposed task is not “early recognition” or “auto-completion,” but a fundamentally different problem centered on modeling future behavioral semantics.

---

> > > > ### Author Response · Authors · 2025-12-03
> > > > **Second Round Response to Reviewer 7Xxa (2/2)**
> > > >
> > > > **On “Systemic Overclaiming” and Narrative Restructuring**
> > > >
> > > > We sincerely thank the reviewer for pointing out the issue of narrative overclaiming. We took this concern seriously and conducted a substantial restructuring across the entire manuscript, rather than cosmetic renaming. Our revisions ensure that the terminology and wording of the paper are fully consistent with "future-aware behavior understanding," and not an overclaim intentional prediction.
> > > >
> > > > Major revisions reflected in the updated PDF (highlighted in the manuscript):
> > > >
> > > > - Title revised to “MoGIC: Boosting Motion Generation via Future-aware Behavior Understanding and Visual Context”
> > > > to accurately reflect the technical scope without implying cognitive intent. (page 1)
> > > >
> > > > - Abstract fully rewritten to emphasize modeling of high-level semantic structures and future behavior patterns that govern how actions unfold, without any reference to intention-like wording. (page 1)
> > > >
> > > > - Second paragraph of the Introduction rewritten to completely remove references to “internal causal structure.” We now state that “a model should also learn the structural dependency between early motion and logically consistent future behaviors.” (page 1)
> > > >
> > > > - Fourth paragraph of the Introduction restructured to clarify the key motivation of future-aware behavior understanding: “The key insight is that early motion segments contain latent cues about how actions are likely to unfold, and such cues must be captured through structured semantic modeling rather than direct language-motion mapping.” (page 2)
> > > >
> > > > - Method section globally realigned to reflect the updated terminology and conceptual framing, including modification in Figure 1, renaming of the module, redefinition of the formulation, and consistent terminology updates throughout Sections 3.1–3.3. (pages 3–6)
> > > >
> > > > - Experiments (Section 5) and Conclusion (Section 6) rewritten to conform to the revised terminology. (pages 7–10)
> > > >
> > > > All modifications are explicitly highlighted in the updated PDF for transparency. We believe that, after these substantial narrative adjustments, the overall framing is now clearer, more rigorous, and fully avoids the overclaiming concerns raised by the reviewer.

---

### Official Review · Reviewer_imB4 · 2025-11-01

**Soundness:** 3
**Presentation:** 3
**Contribution:** 3
**Rating:** 6
**Confidence:** 3

**Summary:**

The paper presents MoGIC, a unified framework for multimodal human motion generation, supported by a large-scale dataset Mo440H , a modular design with vision priors, an intention head, and Mixture-of-Attention, and demonstrates strong performance through extensive experiments, though some aspects like the intention head’s qualitative role could be clarified.

**Strengths:**

1. The Mo440H dataset is large-scale, covering a wide range of interactions and motions, providing a rich and diverse benchmark for training and evaluation.
2. The model’s modular design is well-structured, incorporating vision priors, an intention prediction head, and a Mixture-of-Attention mechanism, which appears reasonable for aligning multimodal signals with motion tokens. This method also demonstrates strong effectiveness, consistently outperforming baselines across multiple tasks and datasets.
3. Extensive experiments, including quantitative metrics and multiple downstream tasks, convincingly demonstrate the effectiveness of the proposed framework.

**Weaknesses:**

1. Ambiguous Definition of Intention: The paper claims that intention modeling is crucial for producing high-quality motion (e.g., caption supervision improves motion quality, but the gains are notably smaller than those from intention prediction). However, it is unclear how “intention” is fundamentally different from the original captions or text. In Figure 2, the intention text appears nearly identical to the motion description, raising the question of whether the Intention Prediction Head (IPH) truly captures high-level latent goals of human motion, or if it simply learns to reconstruct or summarize the original textual input. An additional consideration is whether the IPH is strictly necessary: could the model achieve similar benefits by simply augmenting the original text dataset with intention labels, rather than introducing a separate intention prediction module?
2. Effectiveness of Visual Priors. According to the qualitative results presented in the paper, the introduction of visual input sometimes leads the model to behave more like performing pose reconstruction conditioned on the visuals, rather than effectively leveraging the visual prior to improve the diversity or realism of the generated motions. From the quantitative results, there is currently a lack of clear metrics that convincingly demonstrate that visual input significantly improves the quality of generated motions. Even after language fine-tuning, the supposed benefits of the visual prior, as claimed by the authors, are not evident, and motion diversity does not appear to improve. Therefore, the actual contribution of the visual modality to motion generation performance remains to be further validated.
3. Limited Qualitative Analysis: The paper presents only a limited number of Vision & Language → Motion examples and provides relatively little qualitative analysis of the Intention Prediction Head, making it somewhat unclear how effectively it functions.

**Questions:**

Some minor formatting issues are noticed: missing punctuation(Line 236), inconsistent capitalization (Line 958), and in tables, it would be clearer if key entries were bolded.

---

> ### Author Response · Authors · 2025-11-24
> **Response to Reviewer imB4 (1/3)**
>
> We sincerely thank the reviewers for their constructive feedback. We appreciate the time and effort invested in reading our manuscript and providing valuable suggestions. We have carefully addressed comments below.
>
> > W1: Ambiguous Definition of Intention.
>
> Thank you for pointing out that the current description of “intention” may appear ambiguous. We provide a more precise explanation below.
>
> **(1)	What is the difference between “Intention Prediction” and “Motion Caption”?**
>
> The intention text and the caption text are identical. However, the task definition and the input visibility differ fundamentally.
>
> - In Intention Prediction, the model receives only **the first 50%** of the motion sequence and must output a full textual description of the **entire motion**. This requires the model to **infer future-aware behavior semantics** (expressed in text) that is not present in the visible frames, not only perform text reconstruction.
>
> - In Motion Caption, the model has access to the full motion sequence and only needs to describe what already happened, without any future inference.
>
> Importantly, both Intention Prediction and Motion Caption are functional capabilities of our framework. Intention Prediction is used during cross-modal generative training as one of the training objectives, providing the model with a future behavior-inference signal that shapes its shared latent representation. After this stage, we further fine-tune the same model on the Motion Caption task as one of the downstream applications to demonstrate the extensibility of our architecture.
>
> **(2)	Why this difference matters**
>
> Although the output texts are identical, the goal of these two tasks is entirely different. Inferring missing behavior semantics from partial observations requires the model to understand the structural dependency between early motion and logically consistent future behavior. In another words, Intention Prediction enforces model to understand *“Given what I see so far, what is the person likely trying to do?”* from partial observations, while the Motion Caption enforces descriptive generation (*“what happened in the motion I fully observed?”*).
>
> This distinction is reflected in our ablation results (Table 2).
>
> | ID | Methods                           | Top1            | Top2            | Top3            | FID↓            | Matching↓       | Diversity↑      |
> | -----|-----------------------------------|-----------------|-----------------|-----------------|-----------------|-----------------|-----------------|
> | A   | MoGIC *only L2M loss*      | 0.637   | 0.836   | 0.908   | 0.201   | 2.003   | 12.392  |
> | B   | MoGIC *L2M + Caption loss* | 0.646   | 0.845   | 0.919   | 0.198   | 1.910   | **12.623**  |
> | C   | MoGIC *L2M + Intention loss*  | **0.652** | **0.851** | **0.926** | **0.134** | **1.889** | 12.434  |
>
> We compare:
> - (A) baseline using only L2M loss,
> - (B) adding caption loss on fully observed motions,
> - (C) adding Intention Prediction on partial motions.
>
> Model C substantially outperforms Model B, reducing FID from 0.198 to 0.134, a 32.3% improvement.
>
> This indicates that the performance gain is *not* simply due to “text reconstruction or summarization.” Instead, the improvement primarily arises from the model’s requirement to infer future-aware behavior semantics from incomplete motion. If intention prediction are equivalent to caption reconstruction, its performance would be comparable to caption-loss training, but the 32.3% FID reduction shows that it provides a significantly stronger supervision signal.
>
> **(3) Why we cannot replace IPH with “just augment captions”**
>
> Intention Prediction and Motion Caption are two distinct tasks. As mentioned above, we demonstrated this in Table 2 by replacing Intention Prediction with Motion Caption for our ablation study. The results clearly show that Intention Prediction leads to a significant improvement in motion generation quality.
>
> ***Noting:*** To avoid any potential conceptual ambiguity, we adopt reviewer 7Xxa’s suggestion and replace the term Intention Prediction with **Future-Aware Behavior Understanding**, which more precisely reflects the nature of our method.

---

> ### Author Response · Authors · 2025-11-24
> **Response to Reviewer imB4 (2/3)**
>
> > W2: Effectiveness of Visual Priors.
>
> **(1) Visual Prior vs. Pose Reconstruction (Why it is not reconstruction)**
>
> We would like to clarify that our visual modality is deliberately designed as a "weak prior" rather than a dense 3D supervision signal. Our architecture prevents this by:
>
> - **Sparse & Unaligned Input**: We use 1 fps RGB frames that are not temporally aligned with the 30 fps motion sequences, preventing any frame-to-frame or trajectory-level mapping.
> - **Soft Constraints via Attention**: Visual features are injected solely via the Adaptive Top-k Cross-Attention module, which acts as a semantic guide (object context, interaction type) rather than a geometric template.
> - **Generative Nature**: Motion is generated through sampling in the diffusion latent space. Thus the model cannot copy or regress joint trajectories; the visual inputs can only shape the high-level plausibility of the samples.
>
> These design choices collectively guarantee that the visual modality cannot collapse the task into motion reconstruction.
>
> **(2) The Necessity of Visual Priors for Realism (Qualitative Perspective)**
>
> As shown in Figure 3, a text-only model might generate a motion where a person "lifts a barbell overhead" even when the context implies a different interaction (and it is physically implausible). Even a weak 1 fps visual cue captures interaction semantics and coarse posture evolution, effectively suppressing unrealistic motions while keeping the generation space diverse.
>
> **(3) Quantitative Effectiveness of Vision Modality**
>
> We analyze the effect of the visual modality both during representation learning and as an active condition at test time.
>
> During the Cross-Modal Generative Training phase, the model jointly learns V2M, LV2M, L2M, intention prediction, and partial-motion modeling, followed by an L2M fine-tuning stage. During the L2M fine-tuning phase, the visual input is functionally inactive to optimize the model for text-only inference. Although vision is inactive during L2M fine-tuning, the model that has been exposed to visual signals ("MoGIC w/ L2M FT") achieves better motion realism than the purely text-motion counterpart (FID: 0.134 → 0.123). This shows that visual information helps the model learn a stronger motion manifold even when vision is absent at inference.
>
> We further validate the visual modality as an active condition in both Motion Generation and Motion In-betweening tasks (Table 5):
>
> - **Motion Generation**: When visual input is combined with language (LV2M), the performance is significantly better than the text-only baseline. The FID drops from 0.330 to 0.266, representing a 19.3% improvement in motion quality. This clearly demonstrates the strong contribution of visual conditions in standard generation tasks.
>
> - **Motion In-betweening**: Using visible motion segments plus visual conditions achieves a satisfactory FID of 0.205, which demonstrates the effectiveness of vision as a condition. We note that in this specific task, adding vision to language yields results similar to the language-only condition. This is expected: in motion in-betweening, the visible motion frames already impose spatiotemporal constraints that are ambiguous in text modality. Consequently, the marginal gain of stacking modalities is naturally lower compared to the pure generation task.
>
> **(4) "Weak" Prior Preserves Diversity.**
>
> The reviewer noted that diversity does not seem to improve with visual input. We wish to clarify that maximizing diversity is not the objective of the visual module. If we were to use dense visual signals (e.g., 30 fps video), the task would collapse into a deterministic "video-to-motion" mapping (reconstruction), reducing diversity to near zero. By using a sparse (1 fps) visual prior, we maintain the generative nature of the diffusion model. The model is free to sample from the diverse latent space, while the visual prior anchors the motion to reality. This ensures the model generates diverse but plausible motions, rather than strictly replicating a single deterministic path. As shown by the Diversity metric in Table 5, the diversity of motion generation was still well preserved after the introduction of visual conditions.
>
> In summary, the visual prior is effective not because it forces the model to copy the input, but because it constrains the generative space to eliminate unrealistic hallucinations while preserving the diversity of the diffusion process.

---

> > ### Author Response · Authors · 2025-11-24
> > **Response to Reviewer imB4 (3/3)**
> >
> > > W3: Limited Qualitative Analysis.
> >
> > We thank the reviewer for the suggestion. We will include additional visualization results in the updated PDF and supplementary materials.
> >
> > > Q1: About Formatting Issues.
> >
> > We thank the reviewer for pointing out these issues. We will correct them in the updated PDF.

---

> > > ### Comment · Reviewer_imB4 · 2025-11-26
> > > **Response to the authors**
> > >
> > > Thank you for the clarification. I agree that the term **Future-Aware Behavior Understanding** more accurately reflects the nature of the method.
> > >
> > > I appreciate the revision and will increase my score to accept.

---

> > > > ### Author Response · Authors · 2025-12-03
> > > > **Second Round Response to Reviewer imB4**
> > > >
> > > > Thank you for your valuable feedback. Your comments have been very helpful in optimizing our articles.

---

### Comment · Area_Chair_9AbT · 2025-11-26

Dear reviewers,

The authors have responded. We kindly ask you to review the authors' responses to your comments and provide your feedback. Thank you.

Best,

AC

---

### Author Response · Authors · 2025-12-03
**Declaration of revisions to the paper**

We sincerely thank the reviewers for their detailed and constructive comments. These suggestions have greatly improved the clarity and overall quality of the manuscript. In the revised PDF, we have made substantial updates to both the main paper and the supplementary material, with all modifications clearly highlighted.

**Addressing Reviewer 7Xxa’s concern on overclaiming**

We have replaced all references to intention prediction with the more precise term future-aware behavior understanding, which better reflects the actual motivation and role of this component: to capture the high-level semantic structures and future behavior patterns that govern how actions unfold. Corresponding revisions have been made throughout the paper, including the Abstract, Introduction (paragraphs 2, 4, and 7), Sections 3.2 and 3.3, Figure 1, the training section (Section 4), and the experimental sections (5.2, 5.3, 5.4), as well as the Conclusion, ensuring narrative and conceptual consistency.

**Addressing Reviewer LpQA and Reviewer 1tfi’s concerns on video sampling strategy**

We added a new ablation study on Video Sampling Strategy in Section 5.4 (Table 6), evaluating different sampling rates and sampling methods and their influence on the vision+language-to-motion task.

**Addressing Reviewer 7Xxa and Reviewer 1tfi’s comments on the “arbitrariness” of the 50% prefix definition**

We added an additional experiment in Section 5.4 (Table 7), analyzing how different visible-motion ratios affect the performance of future-aware behavior understanding.

**Addressing Reviewer LpQA’s comment on the distribution of action types**

We added verb-frequency statistics in Supplementary Section A.1, Figure A3.

**Addressing Reviewer 1tfi’s questions about the data processing pipeline**

We added a detailed standardized data processing description in Supplementary Section A.1, covering data filtering, normalization, and motion representation.

**Addressing Reviewer 7Xxa and Reviewer LpQA’s concerns about the reliability of automatically generated labels**

We added a human evaluation study (Table A1) and a zero-shot evaluation on the human-labeled dataset (Table A2) in Supplementary Section A.1.

**Addressing Reviewer LpQA and Reviewer imB4’s comments on missing vision-language-to-motion visualizations**

We added corresponding visualization results and analysis in Supplementary Section A.3 (Figure A6).

**Addressing Reviewer LpQA’s concern on generalization ability**

We added zero-shot evaluations on the IDEA400 dataset (Table A3) and the FineMotion dataset (Table A4) in Supplementary Section A.4.

Additionally, following the reviewers’ suggestions, we refined several parts of the manuscript to more clearly highlight the core contributions of our work. All corresponding changes have been highlighted in the revised PDF.

---

### Meta-Review · Area_Chair_1KCa · 2026-01-03

**Summary:**

The submission reports strong empirical gains for text-conditioned motion generation and introduces a large aggregated tri-modal dataset (Mo440H), with a modular design combining (i) a prefix-to-text auxiliary objective (“intention prediction” / later reframed as “future-aware behavior understanding”), and (ii) sparse visual context conditioning. Reviewers generally acknowledge the performance improvements, but the main concerns center on (a) **conceptual framing and overclaiming**, where the “intention/causal logic” narrative appears to overreach relative to what is implemented (a 50%-prefix-to-text proxy task), and (b) the **arbitrariness/justification of the 50% prefix definition as a core design choice**, which one reviewer argues is not resolved by wording changes alone.

Additional concerns include the **reliability/interpretability of results on an auto-captioned dataset** (Mo440H) and whether improvements might be tied to synthetic label style; and the **unclear incremental benefit of visual priors** (risking “reconstruction-like” behavior and lacking crisp evidence of improvements), plus requests for broader qualitative evidence and clearer data processing details.

**Reviewer Concerns:**

### Conceptual framing / “intention understanding” overclaiming (Reviewer 7Xxa, imB4)
* **Outstanding**
* Authors rename and restructure the paper toward “future-aware behavior understanding,” and add clarifications/ablations contrasting intention-loss vs caption-loss. However, Reviewer 7Xxa explicitly states they remain unconvinced and that the issue is systemic (not only wording), and maintains their rating.

### Arbitrariness / justification of the “50% prefix” definition (Reviewer 7Xxa)
* **Partially addressed (still debated)**
*  Authors add an ablation across prefix ratios and argue 50% performs best (in the revised manuscript). Despite this, the reviewer indicates the rebuttal does not sufficiently justify the conceptual leap tied to “intention/causal logic.”

### Synthetic labels / possible “mimicking” of the VLM label style in Mo440H (Reviewer 7Xxa, LpQA)
* **Largely addressed**
* Authors provide multiple mitigating checks (same labels for baselines; results on human-labeled HumanML3D; human A/B evaluation; and zero-shot tests on a human-labeled dataset). Reviewer 7Xxa notes the added zero-shot validation helps alleviate reliance on synthetic labels (even though they maintain other objections).

### Effectiveness/role of visual priors (Reviewer imB4, 1tfi)
* **Partially Addressed**
* Authors clarify “weak prior” design and add/mention new ablations on video sampling strategy and more visualizations. Still, the case for when/how vision contributes beyond text-only (and how this differs from reconstruction-like conditioning) likely needs stronger evidence and clearer positioning.

### Data processing details / potential leakage & sampling strategy questions (Reviewer 1tfi)
* **Addressed**
*  Authors describe standardization steps and rationale for the setup, including ensuring sampling is done consistently and discussing dataset filtering/standardization.

**Reviewer Scores:**

All reviewers participated actively in the rebuttal discussion. Following the rebuttal, two reviewers explicitly indicated that they would maintain or increase their score (from 6 and 8), while the other two reviewers stated that they would retain their objections and keep a below-threshold recommendation (score 4). Overall, despite meaningful clarifications and additional evidence in the rebuttal, the discussion did not lead to convergence toward acceptance.

In addition, though no reviewer pointed this out, I found no video demonstration was provided for review which could be a critical issue as a work for 3D motion synthesis.

Overall, I do not recommend acceptance of this work.

---

### Decision · Program_Chairs · 2026-01-26

Reject